# `FOCoOp`: Enhancing Out-of-Distribution Robustness in Federated Prompt Learning for Vision-Language Models

Xinting Liao [* 1]   Weiming Liu [* 1]   Jiaming Qian [1]   Pengyang Zhou [1]   Jiahe Xu [1]   Wenjie Wang [2]   Chaochao Chen [1]
Xiaolin Zheng [1]   Tat-Seng Chua [3]

## Abstract

Federated prompt learning (FPL) for vision-language models is a powerful approach to collaboratively adapt models across distributed clients while preserving data privacy. However, existing FPL approaches suffer from a trade-off between performance and robustness, particularly in out-of-distribution (OOD) shifts, limiting their reliability in real-world scenarios. The inherent in-distribution (ID) data heterogeneity among different clients makes it more challenging to maintain this trade-off. To fill this gap, we introduce a `Federated` `OOD-aware` `Context` `Optimization` (`FOCoOp`) framework, which captures diverse distributions among clients using ID global prompts, local prompts, and OOD prompts. Specifically, `FOCoOp` leverages three sets of prompts to create both class-level and distribution-level separations, which adapt to OOD shifts through bi-level distributionally robust optimization. Additionally, `FOCoOp` improves the discrimination consistency among clients, i.e., calibrating global prompts, seemly OOD prompts, and OOD prompts by Semi-unbalanced optimal transport. The extensive experiments on real-world datasets demonstrate that `FOCoOp` effectively captures decentralized heterogeneous distributions and enhances robustness of different OOD shifts. The project is available at GitHub.

## 1. Introduction

In recent days, pretrained vision-language models (VLMs), e.g., CLIP (Radford et al., 2021) and ALIGN (Jia et al., 2021), are widely studied for their benefits of unifying multimodal data and learning transferrable representation across downstream tasks (Liu et al., 2024). To meet the necessity of privacy preservation, federated prompt learning (FPL) methods are recently emerging, which utilize transferable knowledge of VLMs by collaboratively tuning prompt contexts with decentralized data (Guo et al., 2023b; Li et al., 2024). It reduces the overwhelming computation and communication burdens, while bringing effective personalized transferring under privacy regularization. However, the existing FPL methods mainly suffer from a significant trade-off between performance and out-of-distribution (OOD) robustness, i.e., enhancing the accuracy comes at the cost of failing to generalize covariate-shift data or detect semantic-shift data (Lafon et al., 2025; Kumar et al., 2022). As shown in Fig. 1, though the classification accuracies of existing FPL methods (e.g., FedOTP (Li et al., 2024), and Prompt-Folio (Pan et al., 2024)) benefit from maintaining global generalization and local personalization, they almost fail to harness OOD robustness, e.g., detecting OOD samples in downstream tasks. Because OOD samples are completely unseen to the pretraining distribution of the CLIP or the fine-tuning distribution of clients in FPL (Liao et al., 2024b; Yu et al., 2023).

***This brings the problem of enhancing OOD robustness for federated prompt learning on pretrained VLMs.***

Applying federated OOD-aware prompt learning on VLMs suffers from two aspects of challenges, i.e., **CH1:** *How to maintain the class-level and distribution-level separations for distinguishing client data samples?* and **CH2:** *How to enhance the consistency of OOD robustness among all clients?* **The first challenge** arises when applying prompt learning in federated scenarios, leading to inferior class discrimination and reduced distinction between different distributions. Because each client is not only limited to exploit the whole in-distribution (ID) data distribution, but also unavailable for modeling the OOD shifts of data distribution (Liao et al., 2024b). On one thing, it is inevitable for the client model to be overfitting for prompt learning with local data, degrading generalization for unseen ID data in other clients, and covariate-shift data (Cui et al., 2024). On another thing, it brings a mismatch between OOD prompts and real-world OOD distributions, impacting client detection

---

*Equal contribution [1]Zhejiang University [2]University of Science and Technology of China [3]National University of Singapore. Correspondence to: Xiaolin Zheng <xlzheng@zju.edu.cn>.

*Proceedings of the 42nd International Conference on Machine Learning*, Vancouver, Canada. PMLR 267, 2025. Copyright 2025 by the author(s).

Figure 1: The performance and OOD robustness of FPL methods. The FPR95 reflects detection capability, while ACC and CACC are generalization performance on ID data and covariate-shift data, respectively.

capability, as in Fig. 1. **The second challenge** stems from personalization of heterogeneous data, where variations in domains and class representations hinder achieving performance consistency. The inherent heterogeneity of client data prevents the direct application of centralized OOD methods, limiting their effectiveness in improving the OOD robustness of FPL. In Fig. 1, the adaptions of FedAvg (McMahan et al., 2017) with centralized OOD prompt tuning methods fail to effectively handle heterogeneous data, where only FedGalLoP (Lafon et al., 2025) maintains its performance. The crucial reason is that each client maintains the local OOD robustness on heterogeneous data, which is inconsistent among clients in FPL. Even clients in FedGalLop are still confused about identifying data unseen at local but presented in other clients (Yu et al., 2023).

In this work, we promote the OOD robustness of federated prompt learning on VLMs, and propose a **F**ederated **O**OD-aware **C**ontext **Op**timization framework, i.e., FOCoOp. Specifically, FOCoOp integrates the *Bi-level OOD Separations (BOS)* and *Global-view OOD Consistency (GOC)* modules to simultaneously learn global ID prompts for generalization, local prompts for personalization, and OOD prompts for detection. To address **CH1** in each client, **BOS** learns three types of prompts to align local ID images with bi-level distributionally robust optimization, which perturbs global prompts and OOD prompts to explore wider semantic matching with limited image data. The perturbations on global prompts improve class-level separation by penalizing the mismatching scores with ID image data, thus avoiding interference with the original class discrimination among decentralized data. Similarly, **BOS** perturbs OOD prompts to minimize the impact of overall semantic matching between ID image data and OOD prompts, bringing more distinctive distribution-level separation. For tackling **CH2** in server, **GOC** aggregates global prompts from participating clients, and aligns them with OOD prompts via Semi-unbalanced optimal transport mapping. Based on the mapping, **GOC** selects seemly OOD prompts that are mostly close to ID global prompts, to calibrate aggregated global prompts, and sends the most distant OOD prompts back to clients. This approach mitigates ambiguous identification of

OOD prompts for each client, reducing misclassification of data in other clients, and detecting semantically novel OOD data from a global perspective.

To conclude, our main contributions come from four aspects: (1) we are the first to propose a federated OOD-aware prompt learning framework, i.e., FOCoOp, which maintains performance without sacrificing OOD robustness. (2) We design the BOS module, which leverages bi-level distributionally robust optimization to enhance class-matching between images and prompts while ensuring a clear separation between ID and OOD data. (3) We develop the GOC module, which utilizes semi-unbalanced optimal transport mapping to calibrate OOD prompts and global prompts in consistency. (4) In experiments, we validate the effectiveness of FOCoOp on extensive real-world datasets with different tasks, achieving consistently competitive results.

## 2. Related Work

### 2.1. Federated Learning on VLMs

Federated learning (FL) models decentralized data in each client and aggregate client models for a global model in server (McMahan et al., 2017). Personalized federated learning emphasizes tailoring to personalized performance through regularization (Li et al., 2020; Karimireddy et al., 2020), contrastive learning (Li et al., 2021), model decoupling (Chen & Chao, 2021; Dong et al., 2022), hyperbolic modeling (Liao et al., 2023; Liu et al., 2025a), and so on. However, conventional FL methods require the collaboration of client models with full parameter sets on the server, which becomes impractical as model size is growing by scaling law (Li et al., 2024; Cui et al., 2024). Recent works have explored parameter-efficient fine-tuning methods to reduce the communication and computation costs. For example, PFPT (Weng et al., 2024) uses visual prompts to tackle class-imbalance problem in pretrained vision models, and FLoRA (Wang et al., 2024b) utilizes Low-rank adapters with varying ranks to fine-tune language models. Similarly, we further consider using federated prompt learning to collaboratively adapt client data by tunable textual prompts

rather than entire VLMs, reducing communication and computation cost (Guo et al., 2023b;a). Recently, a series of work focuses on balancing personalization and generalization, e.g., CLIP2FL (Shi et al., 2024), DiPrompt (Bai et al., 2024a), FedOTP (Li et al., 2024), FedTPG (Qiu et al., 2023) and pFedPG (Yang et al., 2023). FedFolio (Pan et al., 2024) further provides theoretical insights into trade-offs between generalization and personalization. Although these methods improve federated prompt learning for heterogeneous data, these works overlook the OOD robustness issues.

## 2.2. OOD Robustness in Federated Learning

It is a long-term issue but rarely a topic for improving OOD robustness in federated scenarios (Hendrycks & Gimpel, 2016; Li & Wang, 2024; Huang et al., 2024). Recent FL methods aim to enhance generalization by preserving invariant relationships between data and labels (Jiang & Lin, 2022; Tan et al., 2023; Tang et al., 2022), smoothing local loss landscapes (Qu et al., 2022), and capturing robust representations to handle heterogeneous distributions and adapt to unseen clients (Yuan et al., 2021; Nguyen et al., 2022a; Guo et al., 2023c; Liu et al., 2021). FedGOG (Zhou et al., 2025a) further improves OOD generalization in decentralized graph data (Zhou et al., 2025b). Meanwhile, FOSTER (Yu et al., 2023) learns a class-conditional generator to synthesize virtual external-class OOD samples and facilitate OOD detection in FL for the first time. And FOOGD (Liao et al., 2024b) captures global distribution with score matching model to tackle OOD generalization and detection simultaneously. Nevertheless, these models are not scalable for large pretrained VLMs and struggle to directly adapt to FPL while maintaining OOD robustness.

## 2.3. OOD Robustness on Pretrained VLMs

The pretrained VLMs contain large-scale model parameters and provide transferrable representation for OOD generalization and zero-shot capabilities (Radford et al., 2021; Jia et al., 2021; Liu et al., 2025b). However, VLMs rely heavily on pretrained textual-image matching distribution, causing the degradation of generalization and detection capabilities once the textual prompts are diverse and incorrect (Zhou et al., 2022; Mayilvahanan et al., 2023; Yang et al., 2024; Shu et al., 2023). CoOp (Zhou et al., 2022) proposes to learn the representation vector of prompt context words during adapting pretrained VLMs, enhancing the generalization on distribution shifts. To enhance robustness, OOD generalization and detection are further explored. For instance, CDC (Zhang et al., 2024) employs causal analysis (Wang et al., 2024a; 2025a; Qi et al., 2024) to identify task-irrelevant knowledge interference. Similarly, CLIPN (Wang et al., 2023) finetunes VLMs to generate negative prompts that assess the probability of an OOD concept. Moreover, ID-Like (Bai et al., 2024b) extends pretrained

VLMs to detect OOD data that is highly correlated with ID data. With the constraints of data privacy and heterogeneity of FPL, it is further demanding to efficiently and consistently apply prompt tuning on pretrained VLMs to adapt decentralized data.

## 3. Methodology

### 3.1. Problem Setting

**Federated Prompt Learning Formulation.** As shown in Fig. 2, FPL methods collaboratively adapt pretrained VLMs with decentralized data among a server and $K$ clients by fine-tuning prompts, i.e., $N$ tunable context embeddings $\boldsymbol{t} = \{\boldsymbol{e}_1, \cdots, \boldsymbol{e}_N\}$. The server is responsible for the generalization of prompts in a global view. And each client $k$ focuses on fine-tuning prompts for frozen VLM using local heterogeneous data $\mathcal{D}_k^{\text{ID}} = \{(\boldsymbol{x}_{k,j}, y_{k,j})\}_{j=1}^{N_k}$, where $\boldsymbol{x}_{k,j}$ denotes $j$-th input sample and $y_{k,j}$ is the associated label. For ID image $\boldsymbol{x}_{k,j}$ belonging to class $c$, i.e., $y_{k,j} = c$, the image encoder $\mathcal{I}$ extracts its visual representation, $\boldsymbol{h}_{k,j} = \mathcal{I}_{\boldsymbol{\theta}}(\boldsymbol{x}_{k,j})$. Correspondingly, we compute the textual embeddings for context prompts based on text encoder $\mathcal{T}$, i.e., $\boldsymbol{e}_c = \mathcal{T}_{\boldsymbol{\theta}}(\boldsymbol{t}, \boldsymbol{n}_c)$, where $\boldsymbol{n}_c$ is the class name embedding of $y_{k,j}$, and all prompts are randomly initialized. Then FPL minimizes similarity between image data and prompt corresponding to its label, which is cosine distances, i.e., $s_{\boldsymbol{\theta}}(\boldsymbol{x}_{k,j}, \boldsymbol{t}) = \frac{\mathcal{I}_{\boldsymbol{\theta}}(\boldsymbol{x}_{k,j})\mathcal{T}_{\boldsymbol{\theta}}(\boldsymbol{t}, \boldsymbol{n}_c)}{\|\mathcal{I}_{\boldsymbol{\theta}}(\boldsymbol{x}_{k,j})\|\|\mathcal{T}_{\boldsymbol{\theta}}(\boldsymbol{t}, \boldsymbol{n}_c)\|} = \frac{\boldsymbol{h}_{k,j}\boldsymbol{e}_c}{\|\boldsymbol{h}_{k,j}\|\|\boldsymbol{e}_c\|}$. And we denote $S(\boldsymbol{x}, \boldsymbol{t}) = S(\boldsymbol{h}, \boldsymbol{e}) = \exp\left(s_{\boldsymbol{\theta}}(\boldsymbol{x}, \boldsymbol{t})/\tau\right)$ as similarity score. After model converges, prompts for aggregation in each client $k$ are sent to the server and aggregated by class, e.g., $\boldsymbol{t}_c = \sum_{k=1}^{K} \frac{|\mathcal{D}_{k,c}|}{\sum_k |\mathcal{D}_{k,c}|} \boldsymbol{t}_{k,c}$.

**Objective of FOCoOp.** FOCoOp is a FPL framework that captures heterogeneous client distributions and enhances OOD robustness by using three sets of prompts, i.e., (1) ID global prompts $\boldsymbol{T}^g = \{\boldsymbol{t}_c^g\}_{c=1}^{C}$ of $C$ classes that captures shared ID global distribution, (2) OOD prompts $\boldsymbol{T}^o = \{\boldsymbol{t}_u^o\}_{u=1}^{U}$ that are trained to mismatch with ID data, and (3) ID local prompts $\boldsymbol{T}_k^l = \{\boldsymbol{t}_{k,c}^l\}_{c=1}^{C}$ that captures heterogeneous distribution in each client $k$. In detail, the local prompts $\boldsymbol{T}_k^l$ adapt local data to realize personalization of client $k$. The global prompts $\boldsymbol{T}^g$ enhance the generalization of covariate-shift and heterogeneity in a global view. And the OOD prompts $\boldsymbol{T}^o$ capture the mismatching relationship between ID visual data and OOD textual prompts. The overall objective is to maintain performance as well as enhance OOD robustness:

$$\min \sum_{k=1}^{K} \ell^{\text{G}}(\mathcal{D}_k^{\text{ID}}, \mathcal{D}_k^{\text{ID-C}}, \boldsymbol{T}_k^l, \boldsymbol{T}^g, \boldsymbol{T}^o) + \ell^O(\mathcal{D}^{\text{OOD}}, \boldsymbol{T}_k^l, \boldsymbol{T}^g, \boldsymbol{T}^o),$$

$$(1)$$

where $\ell^G(\cdot)$ is the in-distribution generalization loss on local testing ID data $\mathcal{D}_k^{\text{ID}}$ and covariate-shift data $\mathcal{D}_k^{\text{ID-C}}$, $\ell^O(\cdot)$ is the out-of-distribution detection loss for OOD data $\mathcal{D}^{\text{OOD}}$.

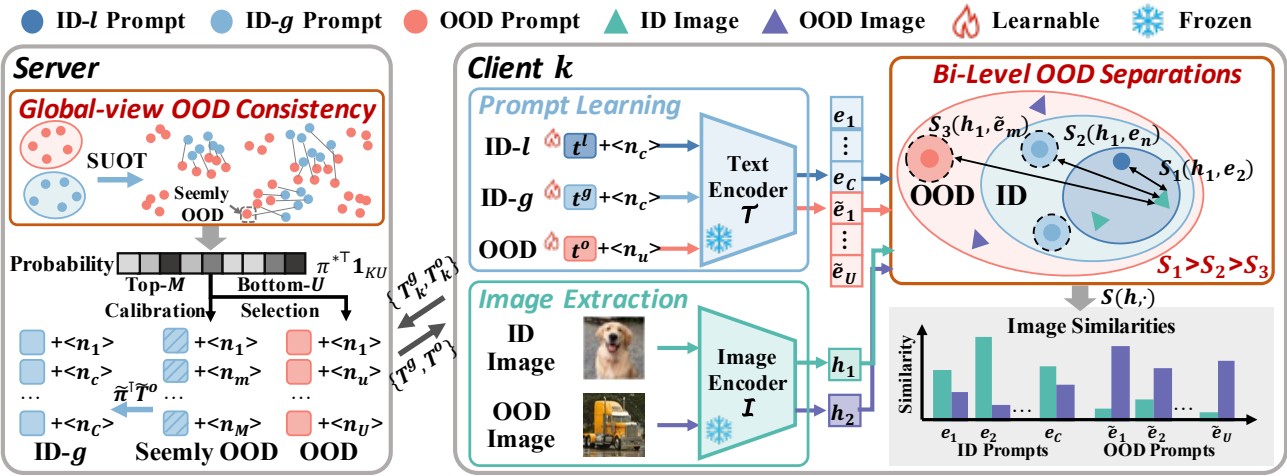

Figure 2: Framework of `FOCoOp`. For each client, `FOCoOp` uses bi-level OOD separations module to fine-tune three sets of prompts adapting to pretrained VLM. While in the server, `FOCoOp` leverages the global-view OOD consistency module to enhance the discrimination among ID global prompts and OOD prompts.

### 3.2. Client: Bi-Level OOD Separations

For the sake of limited access to training data, it mainly includes two aspects of OOD robustness, i.e., OOD generalization and OOD detection, in federated prompt learning for VLMs (Liao et al., 2024b; Bai et al., 2023). Regarding OOD generalization, `FOCoOp` needs to maintain the class-matching between ID prompts and intriguing ID data that are heterogeneous, covariate-shifted, and presented in untrained clients. In terms of OOD detection, `FOCoOp` should identify semantic shifts, rather than mistakenly categorizing unseen samples from other clients as outliers. To tackle the above issues, we first introduce the initialization and modeling procedure of prompt tuning for three sets of prompts. Then we devise bi-level distributionally robust optimization for widely capturing intriguing relationships between different prompts and image data, i.e., (1) class-matching between image and prompts of the same class, and (2) distribution-level separation between local image data and OOD prompts.

**Prompt Context Initialization.** For three sets of prompts, the textual inputs are formulated with class labels $\boldsymbol{t}_c = \{\boldsymbol{t}, \boldsymbol{n}_c\}$, where $\boldsymbol{n}_c$ are the word embeddings corresponding to the $c$-th class name. The global prompts $\boldsymbol{t}^g$ and local prompts $\boldsymbol{t}^l$ share the same ID class set, while OOD prompts $\boldsymbol{t}^o$ are initialized with candidate class names sampled from a lexical database, i.e., WordNet (Fellbaum, 1998), which is widely used in OOD robustness (Jiang et al., 2024; Zhang et al., 2025). Specifically, we calculate the negative cosine similarities between textual embeddings $\boldsymbol{e}_c$ for ID classes $\{y_c\}_{c=1}^C$ and $\widetilde{\boldsymbol{e}}_u$ for OOD candidate classes $\{y_u\}_{u=1}^{\mathbb{U}}$, i.e.,

$$d_u = \text{Percentile}_{\eta \in [0,1]}\left(\left\{-\cos\left(\widetilde{\boldsymbol{e}}_u, \boldsymbol{e}_c\right)\right\}_{c=1}^C\right), \forall_{u \in [\mathbb{U}]}, \quad (2)$$

where $\boldsymbol{t}$ is the fixed context embedding of "a photo the

[CLASS NAME]". Following (Jiang et al., 2024), we select the top $U$ negative class labels $\{\boldsymbol{n}_u\}_{u=1}^U$ based on their distances, i.e., $\text{Top}_U(\{d_u\}_{u=1}^{\mathbb{U}})$.

**Local Prompt Fine-tuning.** After initialization, we compute the similarity scores, and leverage them as prediction probability (Li et al., 2024; Zhou et al., 2022). To realize the class-level separation, we encourage the similarity score of image and ID prompts associated with its label to be high, while hindering remaining similarity scores to be low. The prediction loss can be formulated as:

$$\ell^P = \mathbb{E}_{\boldsymbol{x} \sim \mathcal{D}}\left[-\log(p(y = c|\boldsymbol{x})\right], \quad (3)$$

where $p(y = c|\boldsymbol{x}) = \frac{S(\boldsymbol{x}, \boldsymbol{t}_c)}{\sum_{i=1}^C S(\boldsymbol{x}, \boldsymbol{t}_i) + \sum_{u=1}^U S(\boldsymbol{x}, \boldsymbol{t}_u^o)}$ with $\rho$-proportional ID prompts fusion $\boldsymbol{t}_c = (1 - \rho)\boldsymbol{t}_c^l + \rho\boldsymbol{t}_c^g$. Similarly, we encourage distribution-level separation, by optimizing the loss for prediction probability of all ID prompts that are larger than OOD prompts, i.e.,

$$\ell^D = \mathbb{E}_{\boldsymbol{x} \in \mathcal{D}}\left[-\log p(y_{\text{ID}} = 1|\boldsymbol{x})\right], \quad (4)$$

where $p(y_{\text{ID}} = 1|\boldsymbol{x}) = \frac{\sum_{c=1}^C S(\boldsymbol{x}, \boldsymbol{t}_c)}{\sum_{c=1}^C S(\boldsymbol{x}, \boldsymbol{t}_c) + \sum_{u=1}^U S(\boldsymbol{x}, \boldsymbol{t}_u^o)} = 1 - \frac{\sum_{u=1}^U S(\boldsymbol{x}, \boldsymbol{t}_u^o)}{\sum_{c=1}^C S(\boldsymbol{x}, \boldsymbol{t}_c) + \sum_{u=1}^U S(\boldsymbol{x}, \boldsymbol{t}_u^o)}$. In this case, we obtain the overall objective of constructing class-level and distribution-level separation as below,

$$\mathcal{L}(\boldsymbol{x}, \boldsymbol{\theta}, \boldsymbol{t}^l, \boldsymbol{t}^g, \boldsymbol{t}^o) = \mathbb{E}_{\boldsymbol{x} \in \mathcal{D}}\left[-\log p(y = c|\boldsymbol{x})p(y_{\text{ID}} = 1|\boldsymbol{x})\right]. \quad (5)$$

**Bi-level Prompts distributionally robust optimization.** Though we can achieve the OOD robustness via tuning prompts based on Eq. (5), the separations are intriguing to realize, since the data is heterogeneous and limited in local tuning. To realize the wider distribution exploration for fine-tuning ID local data, we further introduce bi-level

distributionally robust optimization. Specifically, we perturb the global prompts and OOD prompts to the worst-case, whose distribution is mostly divergent from clean prompt distribution by a given discrepancy constraint. Note that global prompts enhance generalization without compromising local personalization. We aim to explore perturbed global prompts within a broad space defined by an optimal transport (OT) divergence from the original global prompts. Unlike KL-divergence, capturing categorical distribution without considering geometry in feature space, OT divergence preserves the geometry of latent feature spaces, which is vital to text-image feature matching in VLMs. While OOD prompts are designed to enhance the detection of open-world semantic shifts unseen during training, they often result in outliers that are geometrically distant from the clean prompt distribution, yet difficult to separate or remove (Wang, 2025). Therefore, OOD prompts are constrained using an unbalanced optimal transport (Liao et al., 2024a; Liu et al., 2024) divergence, simultaneously capturing geometric uncertainty and non-geometric contamination. That is, we seek the worst-case of prompt distributions to match image with textual prompts in a point-to-uncertainty set way, i.e.,

$$\mathcal{L}^{\text{BDRO}} = \inf_{\boldsymbol{\theta}} \sup_{\hat{P}\in\mathbb{P},\hat{Q}\in\mathbb{Q}} \mathbb{E}_{\hat{t}^g\sim\hat{P},\hat{t}^o\sim\hat{Q}} \mathcal{L}(\boldsymbol{x},\boldsymbol{\theta},\boldsymbol{t}^l,\hat{\boldsymbol{t}}^g,\hat{\boldsymbol{t}}^o),$$

$$s.t. \begin{cases} \mathbb{P} = \{P \in \mathbb{D} : D_{\text{OT}}(P, P_0) \leq \eta_1\}, \\ \mathbb{Q} = \{Q \in \mathbb{D} : D_{\text{UOT}}(Q, Q_0) \leq \eta_2\}, \end{cases} \quad (6)$$

where $P_0$ and $Q_0$ are estimated distributions of global prompts $\boldsymbol{T}^g$ and OOD prompts $\boldsymbol{T}^o$, respectively, $\eta_1$ and $\eta_2$ are discrepancy constrains, $D_{\text{OT}}$ and $D_{\text{UOT}}$ are optimal transport distance and unbalanced optimal transport distance defined in Appendix C.1 and C.2, respectively.

**Theorem 3.1.** *Suppose that the optimal dual variables $\tau_1^*$ and $\tau_2^*$ are strictly positive, bi-level separation loss $\mathcal{L}(\boldsymbol{x},\boldsymbol{\theta},\boldsymbol{t}^l,\boldsymbol{t}^g,\boldsymbol{t}^o)$ in Eq. (5) is continuous and differentiable, Bi-level distributionally robust optimization in Eq. (6) can be solved via:*

$$\begin{cases} \hat{\boldsymbol{t}}^g = \arg\min_{\hat{\boldsymbol{t}}^g} \mathbb{E}_{\boldsymbol{t}^g\sim P_0}\left[\sup_{\hat{\boldsymbol{t}}^g\sim\hat{P}} \left\{L(\hat{\boldsymbol{t}}^g) - \tau_1\mathfrak{c}(\hat{\boldsymbol{t}}^g,\boldsymbol{t}^g)\right\}\right], \\ \hat{\boldsymbol{t}}^o = \arg\min_{\hat{\boldsymbol{t}}^o} \mathbb{E}_{\boldsymbol{t}^o\sim Q_0}\left[\sup_{\hat{\boldsymbol{t}}^o\sim\hat{Q}} \left\{L(\hat{\boldsymbol{t}}^o) - \tau_2\mathfrak{c}(\hat{\boldsymbol{t}}^o,\boldsymbol{t}^o)\right\}\right]. \\ \widehat{\mathcal{L}}^{\text{BDRO}} = \arg\min_{\boldsymbol{\theta}} \tau_2\mu\log\mathbb{E}_{\hat{\boldsymbol{t}}^g,\hat{\boldsymbol{t}}^o}\left[\exp(\frac{f(\boldsymbol{\theta})}{\tau_2\mu})\right], \end{cases}$$

$$(7)$$

*where $\mu$ is the regularization of KL divergence, $f(\boldsymbol{\theta}) = \mathcal{L}(\boldsymbol{x},\boldsymbol{\theta},\boldsymbol{t}^l,\hat{\boldsymbol{t}}^g,\hat{\boldsymbol{t}}^o) - \tau_1\mathfrak{c}(\hat{\boldsymbol{t}}^g,\boldsymbol{t}^g) - \tau_2\mathfrak{c}(\hat{\boldsymbol{t}}^o,\boldsymbol{t}^o)$, $L(\boldsymbol{t}^{\{\cdot\}})$ is short for $\mathcal{L}(\boldsymbol{x},\boldsymbol{\theta},\boldsymbol{t}^l,\hat{\boldsymbol{t}}^g,\hat{\boldsymbol{t}}^o)$ optimizing $\boldsymbol{t}^{\{\cdot\}}$, and $\mathfrak{c}(\cdot,\cdot)$ is the cost function of optimal transport, i.e., computing L2-norm distance.*

The proof details are provided in Appendix C.3.

By constructing the class-level and distribution-level separations with bi-level distributionally robust optimization, the local prompt tuning explores wider perception for the generalization via global prompts as well as detection via OOD prompts. The algorithm is presented in Algorithm 2.

### 3.3. Server: Global-view OOD Consistency

Although we can maintain the robustness-aware separations in each client, the data heterogeneity prevents the prompts from optimizing to a consistent optimum for the global distribution. In other words, in each client, global prompts, local prompts, and OOD prompts are fine-tuned by the reference of local data distribution, causing them to be indistinguishable in the server. For example, the OOD prompts in one client match unseen data from other clients with high similarity scores, hurting the generalization in a global view. The seemly OOD prompts are actually ID global prompts to be optimized, bringing the necessity to calibrate the global prompts and OOD prompts in the server. To resolve this ambiguity while preserving generalization performance, we introduce semi-unbalanced optimal transport (SemiUOT) alignment. Specifically, we align the OOD prompts learned among all clients with the global prompts to avoid inconsistent OOD robustness.

**Prompt Aggregation.** The server collects global prompts $\{\boldsymbol{T}_k^g\}_{k=1}^K$ and OOD prompts tuned in clients $\{\boldsymbol{T}_k^o\}_{k=1}^K$. For global prompts of clients, we aggregate them in terms of the corresponding class $\boldsymbol{t}_c^g = \sum_{k=1}^K \frac{|\mathcal{D}_{k,c}|}{\sum_k |\mathcal{D}_{k,c}|}\boldsymbol{t}_{k,c}^g$ as conventional federated methods do (McMahan et al., 2017). However, $\boldsymbol{t}_c^g$ is limited in generalization, since they capture different client data distributions. Moreover, the OOD prompts capture the misalignment between local ID data and prompt context, mistakenly identifying the ID data from other clients as outlier. To enhance the discrimination of global prompts $\boldsymbol{T}^g = \{\boldsymbol{t}_c^g\}_{c=1}^C$ and OOD prompts $\boldsymbol{T}^o$, we seek out the seemly OOD prompts by aligning the distributions of global and OOD prompts using semi-unbalanced optimal transport. We first concatenate $\{\boldsymbol{T}_k^o\}_{k=1}^K$ into $\boldsymbol{T}_{KU}^o$, and apply SemiUOT with $\boldsymbol{T}^g$. The marginal probability of global prompts is constrained to be uniform to avoid bias, while the marginal probability of OOD prompts is loosely constrained to explore wider OOD space. The objective is:

$$\min_{\boldsymbol{\pi}\geq 0} J_{\text{SemiUOT}} = \langle\mathfrak{C},\boldsymbol{\pi}\rangle + \tau\text{KL}(\boldsymbol{\pi}^\top\mathbf{1}_{KU}\|\boldsymbol{a})$$

$$s.t. \quad \boldsymbol{\pi}\mathbf{1}_C = \boldsymbol{b}, \boldsymbol{\pi}\in\mathbb{R}^{C\times KU}, \pi_{cj}\geq 0 \quad \forall_{j\in[KU]}, \quad (8)$$

where the $\boldsymbol{a}$ and $\boldsymbol{b}$ are probability weights initialized equally as $\boldsymbol{a}^\top\mathbf{1}_{KU} = \boldsymbol{b}^\top\mathbf{1}_C$, $\mathfrak{C} \in \mathbb{R}^{C\times KU}$ denotes the cost matrix and it can be calculated via $\mathfrak{C}_{cj} = \|\boldsymbol{t}_c^g - \boldsymbol{t}_j^o\|_2^2 \quad \forall_{j\in[KU],c\in[C]}$. $\text{KL}(\boldsymbol{\pi}^\top\mathbf{1}\|\boldsymbol{a}) = \left[\sum_{j=1}^{KU}\pi_{cj}\log\frac{\sum_{m=1}^{KU}\pi_{cj}}{a_c} - \sum_{m=1}^{KU}\pi_{cj} + a_c\right]$ denotes the KL Divergence between two probability masses $\boldsymbol{\pi}^\top\mathbf{1} \in \mathbb{R}^{KU}$

and $\boldsymbol{a} \in \mathbb{R}^{KU}$. Next, we optimize Eq. (8) via Frank-Wolfe Algorithm (Clarkson, 2010; Jaggi, 2013), which seeks the optimization direction and update it by gradient descent.

**Theorem 3.2.** *Semi-Unbalanced Optimal Transport Optimization of Eq.* (8) *can be solved by Frank-Wolfe Algorithm (Clarkson, 2010; Jaggi, 2013), which iteratively updates* $\boldsymbol{\pi}^{i+1} = \boldsymbol{\pi}^i - \beta(\boldsymbol{\pi}^i - \boldsymbol{s}^i)$, *with step size* $\beta = \frac{1}{i+2}$ *following Armijo condition (Armijo, 1966) and optimal direction* $\boldsymbol{s}^i$ *satisfying:*

$$s^i_{cj} = \begin{cases} b^i_j, & if \quad c^i = \arg\min_c \nabla J_{\text{SemiUOT}}\left(\boldsymbol{\pi}^i_{\cdot j}\right) \\ 0, & otherwise \end{cases},$$

$$(9)$$

*with initialization* $s^0_{cj} = \begin{cases} b^0_j, & c = 0 \\ 0, & otherwise \end{cases}.$

We provide the details in Appendix C.4. The optimal mapping matrix $\boldsymbol{\pi}^*$ is used to figure out the seemly OOD prompts and enhance the OOD prompts in a global consistency.

**Prompt Calibration.** Finally, we seek seemly OOD prompts $\widetilde{\boldsymbol{T}}^o$ and mapping $\widetilde{\boldsymbol{\pi}}$ based on the top-M OOD prompts whose probabilities satisfy $\text{Top}_M(\boldsymbol{\pi}^{*\top}\mathbf{1}_{KU})$. Then we transport it to the semantic space of global prompts, and update the global prompts via exponential moving weight updating.

$$\boldsymbol{T}^g = \alpha \boldsymbol{T}^g + (1-\alpha)\widetilde{\boldsymbol{\pi}}^\top\widetilde{\boldsymbol{T}}^o. \tag{10}$$

While the remaining OOD prompts are filtered to keep top $U$ OOD prompts with less signigicant alignment potential, i.e., $\text{Top}_U(-\boldsymbol{\pi}^{*\top}\mathbf{1}_{KU})$. The final OOD prompts for global updating are reformulated as

$$\boldsymbol{T}^o = \boldsymbol{T}^o_U = \boldsymbol{T}^o_{UK}[\text{Top}_U(-\boldsymbol{\pi}^{*\top}\mathbf{1}_{KU})]. \tag{11}$$

Since the updated OOD prompts are mostly distant from ID global prompts, we enhance the discrimination between the ID distribution and OOD distribution.

In each communication round, the server sends the updated global prompts $\boldsymbol{T}^g$ and OOD prompts $\boldsymbol{T}^o$ as the consistent initial points for local adapting. We finally have communication of federated OOD-aware prompt learning, whose overall procedure is in Algorithm 1. By jointly utilizing Bi-level OOD robustness separation in local client modeling based on Algorithm 2, and global-view OOD robustness consistency in Algorithm 3, FOCoOp simultaneously resolves the trade-off between performance and OOD robustness.

## 4. Experiments

### 4.1. Experimental Setup

**Datasets.** We study the OOD robustness of federated prompt learning on fifteen datasets. **In terms of main-**

**taining performance and OOD robustness**, we simulate heterogeneous distribution following both Dirichlet and Pathological settings (McMahan et al., 2017; Li et al., 2020) on CIFAR-100 (Krizhevsky et al., 2009) and TinyImageNet (Le & Yang, 2015) as conventional work does (Liao et al., 2024b). We test the generalization based on CIFAR-100-C (Hendrycks & Dietterich, 2018) and TinyImageNet-C (Le & Yang, 2015). Meanwhile, we compute on iNaturalist (Van Horn et al., 2018), iSUN (Xiao et al., 2010), Places (Zhou et al., 2017), and Texture (Cimpoi et al., 2014b) for evaluating the OOD detection capability in the testing phase, following existing CLIP-based methods (Wang et al., 2023; Miyai et al., 2024). **To widely evaluate OOD generalization and detection**, we follow previous work of federated prompt learning (Cui et al., 2024; Guo et al., 2023b;a), to study (1) heterogeneous label shift generalization on Food101 (Bossard et al., 2014), DTD (Cimpoi et al., 2014a), Caltech101 (Fei-Fei et al., 2004), Flowers (Nilsback & Zisserman, 2008), and OxfordPet (Parkhi et al., 2012) to predict the accuracy of personalization following pathological heterogeneity, and (2) feature shift domain generalization on DomainNet (Peng et al., 2019), and Office-Caltech10 (Gong et al., 2012), by leave-one-domain-out validation strategy (Nguyen et al., 2022b). Specifically, for $N-1$ domains of one dataset, we train each client with distinct domain data, and test its model generalization on the whole target data of remaining one domain.

**Comparison Methods.** We categorize the comparison methods into two types, i.e., (1) **Existing federated prompt learning methods**: PromptFL (Guo et al., 2023b), FedOTP (Li et al., 2024), FedPGP (Cui et al., 2024), Prompt-Folio (Pan et al., 2024), (2) **Adapting existing centralized OOD robustness methods for federated scenarios**: Fed-LAPT (Zhang et al., 2025), FedGalLoP (Lafon et al., 2025), FedLoCoOp (Miyai et al., 2024). The method implementation details are illustrated in Appendix D.

**Implementation Details and Evaluation Metrics.** We conduct experiments on ViT-B/16 (Dosovitskiy, 2020) CLIP models. To study the heterogeneity generalization on CIFAR-100/TinyImageNet datasets, we simulate both cross-device and cross-silo scenarios. That is, we set local training epoch $E = 2$, communication round $T = 25$, and the number of clients $K = 10$ for fully participation. While in cross-device setting, we choose local training epochs $E = 2$, communication rounds $T = 100$, and $K = 100$ for $10\%$ participation. To obtain fair comparisons, all comparison methods are tuned for converging using their best hyper-parameters, and we report the average of the results from three random seeds. We set the learnable prompt vectors with length as 16, embedding size as 512, class token position as 'end', and random initialization. We choose 1 prompt per class for both local and global ID prompts, and 100 OOD prompts in total. We report the average Top-1 accuracies

Table 1: Main results of federated prompt learning on CIFAR-100 and TinyImageNet.

| Heterogeneity | Pathological Non-overlap ($K = 10$) | | | | | | | | Pathological Overlap ($K = 100$) | | | | | | | |
|---|---|---|---|---|---|---|---|---|---|---|---|---|---|---|---|---|
| Datasets | CIFAR-100 | | | | TinyImageNet | | | | CIFAR-100 | | | | TinyImageNet | | | |
| Methods | ACC | CACC | FPR95 | AUROC | ACC | CACC | FPR95 | AUROC | ACC | CACC | FPR95 | AUROC | ACC | CACC | FPR95 | AUROC |
| PromptFL | 69.35 | 65.14 | 84.51 | 68.28 | 65.58 | 59.37 | 76.75 | 69.82 | 72.86 | 68.98 | 82.07 | 70.92 | 70.76 | 65.36 | 72.51 | 73.38 |
| FedOTP | 90.68 | 88.73 | 38.22 | 87.56 | 82.40 | 78.68 | 57.12 | 79.18 | 89.76 | 78.01 | 51.05 | 85.89 | 74.48 | 71.29 | 66.61 | 73.09 |
| FedPGP | 85.78 | 82.35 | 51.57 | 84.68 | 81.85 | 76.29 | 50.86 | 83.05 | 72.06 | 67.74 | 83.25 | 71.83 | 68.91 | 63.76 | 72.50 | 73.47 |
| PromptFolio | 91.82 | 89.60 | 44.26 | 88.06 | 88.03 | 83.39 | 42.84 | 87.34 | 81.99 | 75.86 | 73.53 | 75.71 | 78.16 | 73.27 | 61.08 | 79.66 |
| FedLoCoOp | 64.13 | 60.23 | 77.59 | 68.76 | 58.08 | 52.05 | 72.96 | 72.07 | 72.93 | 67.98 | 79.76 | 70.54 | 69.05 | 63.22 | 67.92 | 75.06 |
| FedGalLoP | 91.19 | 88.63 | 41.45 | 89.64 | 87.18 | 82.55 | 45.54 | 86.53 | 91.37 | 81.70 | 57.78 | 87.42 | 82.92 | 78.78 | 53.38 | 83.71 |
| FedLAPT | 60.35 | 56.73 | 82.44 | 67.51 | 60.67 | 55.79 | 73.46 | 71.05 | 60.42 | 57.05 | 83.42 | 68.77 | 60.76 | 56.17 | 74.06 | 70.49 |
| FOCoOp | **93.85** | **91.47** | **19.50** | **95.42** | **88.35** | **83.56** | **21.73** | **96.56** | **94.10** | **82.23** | 24.00 | **92.82** | **83.53** | **79.20** | **38.05** | **89.85** |
| -w/o-BOS | 91.07 | 89.06 | 27.31 | 93.08 | 84.39 | 78.16 | 31.09 | 92.60 | 91.35 | 77.45 | 32.85 | 92.30 | 79.83 | 72.25 | 46.30 | 86.30 |
| -w/o-GOC | 88.04 | 85.09 | 37.01 | 87.47 | 85.69 | 80.29 | 28.48 | 93.10 | 89.10 | 74.05 | 41.85 | 88.08 | 72.20 | 66.33 | 50.13 | 86.07 |

Table 2: Main results of federated prompt learning on CIFAR-100 with different Dirichlet distributions.

| Heterogeneity | $\alpha = 0.1$ | | | | $\alpha = 0.5$ | | | | $\alpha = 5.0$ | | | |
|---|---|---|---|---|---|---|---|---|---|---|---|---|
| Methods | ACC | CACC | FPR95 | AUROC | ACC | CACC | FPR95 | AUROC | ACC | CACC | FPR95 | AUROC |
| PromptFL | 71.22 | 67.55 | 76.58 | 72.20 | 75.65 | 71.52 | 82.13 | 69.65 | 74.92 | 71.37 | 79.52 | 74.25 |
| FedOTP | 76.81 | 73.50 | 61.88 | 79.14 | 68.43 | 65.67 | 73.78 | 73.45 | 66.20 | 63.16 | 77.73 | 71.15 |
| FedPGP | 76.77 | 72.55 | 74.81 | 74.45 | 72.95 | 69.25 | 83.65 | 71.37 | 73.01 | 69.15 | 82.57 | 72.65 |
| PromptFolio | 80.07 | 76.89 | 65.30 | 77.95 | 75.98 | 71.98 | 78.61 | 71.44 | 74.19 | 70.60 | 79.64 | 72.74 |
| FedLoCoOp | 67.87 | 63.70 | 76.81 | 70.40 | 74.44 | 70.35 | 73.28 | 72.56 | 74.87 | 70.98 | 74.82 | 73.72 |
| FedGalLoP | 80.53 | 77.61 | 60.72 | 82.66 | 75.87 | 72.85 | 68.72 | 79.66 | 74.32 | 71.14 | 72.72 | 79.13 |
| FedLAPT | 61.20 | 57.54 | 80.28 | 69.97 | 59.41 | 56.33 | 81.97 | 66.73 | 60.03 | 56.29 | 80.13 | 68.42 |
| FOCoOp | **82.42** | **78.52** | **46.56** | **86.98** | **77.71** | **73.59** | **54.26** | **83.40** | **77.66** | **73.59** | **51.02** | **83.22** |
| -w/o-BOS | 79.18 | 76.04 | 54.30 | 82.34 | 74.39 | 70.55 | 58.40 | 81.27 | 75.09 | 71.76 | 54.92 | 81.64 |
| -w/o-GOC | 78.66 | 75.99 | 53.97 | 82.56 | 74.78 | 70.88 | 57.83 | 81.55 | 75.04 | 71.50 | 55.20 | 81.85 |

for generalization of ID (ACC↑) and ID-C (CACC↑). We compute maximum concept matching (MCM) (Ming et al., 2022) as OOD detection score, which is based on similarity between textual features and image features. Based on MCM, we report the standard metrics used for OOD detection, i.e., AUROC (↑) and FPR95 (↓) (Yang et al., 2024).

### 4.2. Performance Evaluation

**Maintaining performance and OOD robustness on heterogeneous data.** We study the capability of Maintaining performance and OOD robustness on heterogeneous data on CIFAR-100, TinyImageNet, and additional five datasets on Tab. 1, Tab. 2, and Tab. 3. Without specification, we use brightness covariate shift as ID-C and texture as OOD data. Since DTD and Texture are identical, and Texture is commonly used for generalization in FPL methods, we evaluate its detection performance by using iSUN as the OOD dataset. **Firstly, existing FPL methods are not OOD-aware.** PromptFL does not perform well because it does not adapt to heterogeneity. The FPL methods considering heterogeneity, e.g., PromptFolio, achieve better results for both generalization ID data and ID-C data, but fail significantly in detecting OOD data. This indicates that these methods can maintain the class-level separation among clients while ignoring the distribution-level separation between ID and OOD data. **Secondly, existing OOD-aware methods are hindered by heterogeneous data.** FedLoCoOp and

FedLAPT perform worse in both generalization and detection, meaning that they cannot robustly capture semantic-matching between image data and contextual prompts with a few of samples. FedGalLoP can achieve OOD robustness, even performing best on OxfordPets. However, it still lacks of discrimination between ID and OOD data, due to locally identifying OOD samples without a global consistency of discrimination. **Thirdly, FOCoOp maintains the performance and OOD robustness.** FOCoOp outperforms other baselines on different heterogeneities and different participation settings. Specifically, FOCoOp detects OOD effectively without sacrificing the prediction performance, validating that class-level separation and distribution-level separation are supposed to be considered at the same time. With the benefits of global-view discrimination between ID global prompts and OOD prompts, the bi-level separations become more consistent and powerful among clients, achieving the best results.

**Ablation Studies.** We design two variants of FOCoOp by removing bi-level OOD separations (-w/o-BOS), and global-view OOD consistency (-w/o-GOC), respectively, to verify the effects of different modules. Though both variants suffer from performance degradation compared with FOCoOp, their detection capability remains competitive. This means that both BOS and GOC play a crucial role in distinguishing semantic-shift data, either considering local distribution-level separation or enhancing prompt discrimination in a

Table 3: Other datasets comparison (%) on the Pathological heterogeneity setting ($K = 10$).

| | Food101 | | DTD | | Caltech101 | | Flowers102 | | OxfordPets | |
|---|---|---|---|---|---|---|---|---|---|---|
| | ACC | FPR95 | ACC | FPR95 | ACC | FPR95 | ACC | FPR95 | ACC | FPR95 |
| PromptFL | 11.87 | 83.16 | 57.11 | 83.47 | 72.83 | 44.18 | 24.03 | 82.25 | 46.23 | 59.88 |
| FedOTP | 51.88 | 73.05 | 91.60 | 25.07 | 66.46 | 50.43 | 84.38 | 40.12 | 88.52 | 37.34 |
| FedPGP | 42.91 | 70.13 | 76.02 | 13.60 | 67.95 | 54.79 | 74.11 | 53.60 | 62.44 | 50.38 |
| PromptFolio | 49.94 | 64.39 | 90.37 | 20.86 | 74.72 | 43.53 | 77.73 | 41.42 | 88.03 | 21.83 |
| FedLoCoOp | 14.25 | 87.97 | 56.25 | 39.21 | 69.67 | 43.79 | 46.86 | 69.25 | 35.89 | 85.42 |
| FedGalLoP | 94.58 | 12.11 | 91.15 | 17.57 | 86.48 | 31.09 | 96.58 | 18.07 | 99.09 | 1.70 |
| FedLAPT | 9.09 | 87.53 | 39.26 | 19.36 | 53.84 | 54.13 | 7.07 | 84.80 | 33.10 | 74.92 |
| FOCoOp | **95.52** | **5.19** | **95.23** | 13.29 | **87.89** | 14.22 | **98.50** | **1.86** | 98.74 | 5.52 |
| -w/o-BOS | 72.55 | 27.02 | 93.29 | 22.61 | 79.44 | 22.14 | 98.17 | 1.94 | 97.66 | 7.47 |
| -w/o-GOC | 78.40 | 19.59 | 94.51 | 20.13 | 79.41 | 24.27 | 98.09 | 2.59 | 98.16 | 6.14 |

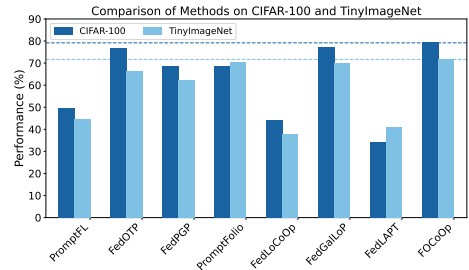

Figure 3: The average generalization results on data with different covariate-shifts.

Table 4: Domain generalization on DomainNet (left) and Office (right). The column notations are short for domain names.

| Method | C | I | P | Q | R | S | Avg (DN) | A | Ca | D | W | Avg (O) |
|---|---|---|---|---|---|---|---|---|---|---|---|---|
| PromptFL | 96.28 | 74.84 | 95.81 | 60.28 | 96.77 | 96.36 | 86.72 | 96.21 | 94.64 | 99.20 | 97.03 | 96.77 |
| FedOTP | 91.03 | 61.52 | 86.98 | 53.04 | 91.16 | 89.73 | 78.91 | 94.64 | 93.15 | 98.93 | 96.46 | 95.80 |
| FedPGP | 93.67 | 75.07 | 93.62 | 58.09 | 95.44 | 95.48 | 85.23 | 95.17 | 95.36 | **99.73** | 97.45 | 96.93 |
| PromptFolio | 95.24 | 75.64 | 94.78 | 59.02 | 95.58 | 95.41 | 85.95 | 96.73 | 94.29 | 98.66 | 97.59 | 96.82 |
| FedLoCoOp | 95.34 | 72.31 | 92.78 | 60.11 | 96.07 | 96.12 | 85.46 | 96.47 | 94.15 | 93.60 | 95.33 | 94.89 |
| FedGalLoP | 95.62 | 75.40 | 94.78 | **65.08** | 96.23 | **96.71** | 87.30 | 97.30 | 96.33 | **99.73** | **98.58** | 97.99 |
| FedLAPT | 92.36 | 66.54 | 89.02 | 48.38 | 94.07 | 92.07 | 80.41 | 77.28 | 84.61 | 86.40 | 86.86 | 83.79 |
| FOCoOp | **96.44** | **76.59** | **96.72** | 62.99 | **97.16** | 96.22 | **87.68** | **98.21** | **96.54** | 99.71 | 98.20 | **98.16** |

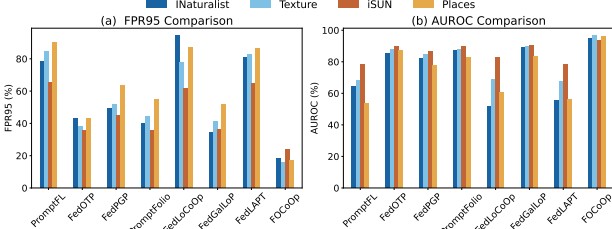

Figure 4: Detection comparison on CIFAR-100.

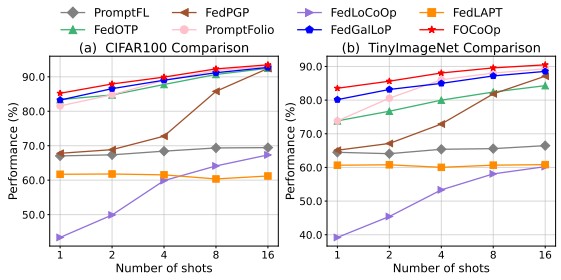

Figure 5: The effect of different number of shots.

global view. FOCoOp-w/o-GOC has a more significant performance drop, illustrating that simply applying bi-level distribution robustness optimization is inferior in heterogeneous data. This also verified the importance of maintaining consistency of ID global prompts and OOD prompts.

**Visualization.** In Fig. 6, we model FOCoOp on Cifar10 (10 ID prompts and OOD prompts, respectively), and sample 100 images per class to compute the average of similarities between images and prompts. The diagonal of the ID

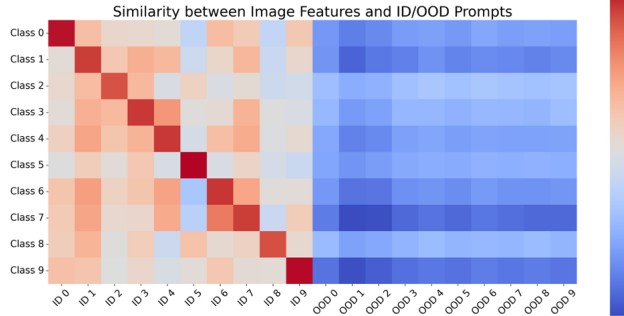

Figure 6: Similarity heatmap of FOCoOp.

prompt matrix shows the highest similarities, suggesting intra-class alignment and clear class separation. Meanwhile, the similarities of OOD prompts are notably lower than those of ID prompts, further indicating clear distribution separation.

**Domain Generalization.** We study the domain generalization on DomainNet and Office in Tab. 4. FOCoOp achieves state-of-the-art domain generalization, yet FedGalLoP and PromptFolio also perform competitively, indicating that all methods effectively leverage the transferability of pretrained VLMs for feature-shift distributions. Compared with label-shifts, heterogeneity impact slightly on all methods, making PromptFL performs well in domain generalization. Among the methods evaluated, FedLAPT exhibits the lowest generalization performance, suggesting that FedLAPT is less effective in leveraging domain-agnostic features.

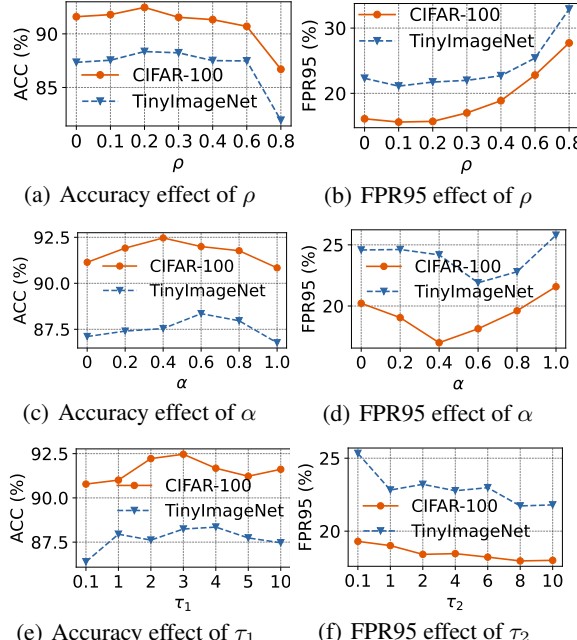

Figure 7: Hyperparameter sensitivity studies.

b), coefficient of updating global prompts $\alpha = \{0, 0.2, 0.4, 0.6, 0.8, 1.0\}$ in Fig. 7(c-d), BDRO hyperparameters $\tau_1 = \{0.1, 1, 2, 3, 4, 5, 10\}$ in Fig. 7(e), and $\tau_2 = \{0.1, 1, 2, 4, 6, 8, 10\}$ in Fig. 7(f) on ACC and FPR95 across CIFAR-100 and TinyImageNet datasets. We substitute ViT-B/16 with ResNet50 (He et al., 2016) to verify in Appendix Tab. 5. We can find that: (1) FOCoOp outperforms in all cases, and we select 8 shots per class for prompt learning as it approximates convergence. (2) Increasing $\rho$ initially improves accuracy but degrades performance beyond an optimal threshold $\rho = 2$, which reflects the same behavior in FPR95. (3) $\alpha$ influences the calibration of ID global prompts with seemly OOD prompts, where different datasets have different turning points. (4) The effects of $\tau_1$ and $\tau_2$ are different, i.e., $\tau_1$ can find the trade-off points where the optimal transport regularization is the best for classification, while $\tau_2$ increasingly encourages uncertainty exploration to have better detection as the value grows. (5) The results validate that FOCoOp maintains strong generalization and detection capabilities even with smaller models.

## 5. Conclusion

In this work, we propose FOCoOp, a Federated OOD-aware Context Optimization framework to enhance robustness and performance in Federated Prompt Learning for VLMs. FOCoOp integrates bi-level OOD separations to improve class-matching and distribution separations, and global-view OOD consistency to align and calibrate global and OOD prompts via semi-unbalanced optimal transport. Extensive experiments across fifteen datasets demonstrate that FOCoOp effectively handles heterogeneous distributions, improves OOD detection, and achieves state-of-the-art performance without compromising generalization, making it a promising solution for federated vision-language model learning.

## Acknowledgement

This research was supported by Zhejiang Provincial Natural Science Foundation of China under Grant No. LZYQ25F020002, and the National Natural Science Foundation of China No. 62172362.

## Impact Statement

This paper provides new insight into maintaining performance as well as enhancing OOD robustness for federated prompt learning on pretrained vision-language models. This aligns with the development of trustworthy machine learning in privacy and robustness. We propose FOCoOp are comprehensively studied with extensive real-world datasets, validating its effectiveness in both performance and robustness.

**Method capability on other datasets.** To comprehensively evaluate the OOD robustness of FPL methods, we present the average ID-C covariate shift generalization in Fig. 3 and OOD semantic shift detection in Fig. 4 and Appendix Fig. 9. Additionally, we provide detailed numerical results in Tables 11–14 Appendix. FOCoOp demonstrates the strongest generalization and detection performance, consistently achieving the highest accuracy and OOD robustness across different covariate-shifts and semantic-shifts on CIFAR-100 and TinyImageNet. FedGaLLoP performs well in ID-C generalization, while suffering from detection due to lacking consistent discrimination among clients. FedOTP and FedPGP strike a balance between generalization and robustness. In contrast, FedLAPT and PromptFL exhibit significant performance degradation, hindering their ability to consistently align image and contextual prompts while compromising OOD robustness.

Table 5: Pathological Non-overlap (10 clients, K=10)

| Dataset | CIFAR-100 | | | | TinyImageNet | | | |
|---|---|---|---|---|---|---|---|---|
| Method(%) | ACC | CACC | FPR95 | AUROC | ACC | CACC | FPR95 | AUROC |
| PromptFolio | 51.96 | 47.47 | 50.14 | 86.45 | 44.70 | 30.64 | 57.57 | 83.03 |
| FedOTP | 55.34 | 50.98 | 61.38 | 74.92 | 43.61 | 28.98 | 72.67 | 73.86 |
| FedLoCoOp | 17.03 | 12.09 | 93.36 | 52.66 | 9.44 | 4.96 | 93.63 | 56.40 |
| FOCoOp | 60.94 | 55.92 | 28.83 | 95.54 | 49.96 | 35.89 | 53.51 | 87.75 |

**Sensitivity Analysis of Hyperparameters.** We present a comprehensive hyperparameter sensitivity analysis, which examines the effect of shot numbers in $\{1, 2, 4, 8, 16\}$ in Fig. 5, effect of ID global and local prompt fusion $\rho = \{0, 0.1, 0.2, 0.3, 0.4, 0.5, 0.6, 0.8\}$ in Fig. 7(a-

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

The supplemental materials consist of five sections: (A) related work details, (B) algorithms, (C) theoretical analysis of optimization, (D) datasets and implementation, and (E) additional experimental results.

## A. Related Work

### A.1. Federated Learning on VLMs

Federated learning (FL) models decentralized data in each client and aggregate client models for a global model at server (McMahan et al., 2017). Personalized federated learning emphasizes tailoring to diverse client data to preserve personalized performance through techniques such as regularization (Li et al., 2020; Karimireddy et al., 2020), contrastive learning (Li et al., 2021), and the decoupling of model parameters (Chen & Chao, 2021; Dong et al., 2022), among others. However, conventional personalized federated learning methods requires the collaboration of client models with full parameter sets on the server, which becomes impractical as model size grows due to scaling laws (Li et al., 2024; Cui et al., 2024). Federated prompt learning collaboratively adapts client data by tunable prompts rather than entire VLMs, which not only utilizes the generalization ability of pre-trained VLMs like CLIP to learn transferable representations, but also preserves data privacy through federated learning (Li et al., 2024; Guo et al., 2023a). PromptFL (Guo et al., 2023b) learns a unified prompt for all clients to enable federated learning. CLIP2FL (Shi et al., 2024) bridges server and client communication using CLIP, handling heterogeneous and long-tailed data. FedPR (Feng et al., 2023) designs federated visual prompts for domain-specific tasks like MRI reconstruction. FedOTP (Li et al., 2024) balances global alignment and personalization with an optimal transport optimization-based approach. FedTPG (Qiu et al., 2023) and pFedPG (Yang et al., 2023) enhance generalization through prompt generation techniques. pFedPrompt (Guo et al., 2023a) adapts prompts for personalized federated learning. FedPGP (Cui et al., 2024) balances generalization and personalization via low-rank decomposition and CLIP guidance. FedFolio (Pan et al., 2024) further provides theoretical insights into trade-offs between generalization and personalization. While these methods improve federated prompt learning, these works overlook the OOD robustness issues, suffering from the trade-off between performance and talking OOD shifts.

### A.2. OOD Robustness in Federated Learning

OOD robustness indicates the capability of model to discriminate distribution shifts, e.g., performance generalization for covariate shifts and outlier detection for semantic shifts (Hendrycks & Gimpel, 2016; Li & Wang, 2024; Huang et al., 2024), which is a long-term issue but seldom studied in federated scenarios. Recent FL methods aim to enhance generalization by preserving invariant relationships between data and labels (Jiang & Lin, 2022; Tan et al., 2023; Tang et al., 2022), smoothing local loss landscapes (Qu et al., 2022), and capturing robust representations to handle heterogeneous distributions and adapt to unseen clients (Yuan et al., 2021; Nguyen et al., 2022a; Guo et al., 2023c; Liu et al., 2021). Meanwhile, FOSTER (Yu et al., 2023) learns a class-conditional generator to synthesize virtual external-class OoD samples and facilitate OOD detection in FL for the first time. And FOOGD (Liao et al., 2024b) captures global distribution with score matching model, and simultaneously tackles OOD generalization and detection based on score function values. Nevertheless, these models are not scalable for large pretrained VLMs and fail to adapt the federated prompt learning to enhance OOD robustness for VLMs.

### A.3. OOD Robustness on Pretrained VLMs

The pretrained VLMs contain large-scale model parameters and provide transferrable representation for OOD generalization and zero-shot capabilities (Radford et al., 2021; Jia et al., 2021). However, VLMs rely heavily on pretrained textual-image matching distribution, causing the degradation of generalization and detection capabilities once the textual prompts are diverse and incorrect (Zhou et al., 2022; Mayilvahanan et al., 2023; Yang et al., 2024). CoOp (Zhou et al., 2022) proposes to learn the representation vector of prompt context words during adapting pretrained VLMs, enhancing the generalization on distribution shifts. Motivated by this, CLIPN (Wang et al., 2023) fineunes VLMs to generate negative prompts that access the probability of an OOD concept. Moreover, ID-Like (Bai et al., 2024b) extends pretrained VLMs to detect OOD data that are highly correlated with ID data. With the constraints of private data and data heterogeneity of FPL, it is further demanding to efficiently and consistently apply prompt tuning on pretrained VLMs to adapt decentralized data. This problem is still unresolved in existing work, since they cannot adapt decentralized data and detect OOD data in a global view.

## B. Algorithm

The overall algorithm of FOCoOp is in Algorithm 1. The steps 1-12 are main procedure of communication server and clients with prompts. In step 9, clients execute local training to improve prompt learning and build bi-level OOD separations, where the details are illustrated in steps 14-24. And the server will calibrate global prompts and OOD prompts in a global view in step 11, which enhances discriminations. Additionally, we provide the crucial algorithms for bi-level OOD separations and global-view OOD consistency as follows. By jointly utilizing Bi-level OOD robustness separation in local client modeling based on Algorithm 2, and global-view OOD robustness consistency in Algorithm 3, FOCoOp resolves the trade-off between performance and OOD robustness, at the same time.

---

**Algorithm 1** Training procedure of FOCoOp

---

**Input**: Batch size $B$, communication rounds $T$, number of clients $K$, local steps $E$, dataset $\mathcal{D} = \cup_{k \in [K]} \mathcal{D}_k$
**Output**: context prompts, i.e., $\boldsymbol{T}_T^g, \{\boldsymbol{T}_{k,T}^l\}_{k=1}^K$, and $\boldsymbol{T}_T^o$

1: **Server executes():**
2: Initialize prompts $\boldsymbol{T}_0^g, \{\boldsymbol{T}_{k,0}^l\}_{k=1}^K$, and $\boldsymbol{T}_0^o$ with random distribution
3: **for** $t = 0, 1, \ldots, T-1$ **do**
4:     **for** each client $k = 1$ to $K$ in parallel **do**
5:         **if** $t == 0$ **then**
6:             Send $\{\boldsymbol{T}_{k,0}^l\}$ to client $k$
7:         **end if**
8:         Send prompts $\boldsymbol{T}_t^g$ and $\boldsymbol{T}_t^o$ to client $k$
9:         $\{\boldsymbol{T}_{k,t}^g, \boldsymbol{T}_{k,t}^o\} \leftarrow$ **Client executes**$(k, \{\boldsymbol{T}_t^g, \boldsymbol{T}_t^o\})$
10:     **end for**
11:     Calibrate prompts $\{\boldsymbol{T}_{t+1}^g, \boldsymbol{T}_{t+1}^o\}$ to achieve global OOD consistency by Algorithm 3
12: **end for**
13: **return** $\{\boldsymbol{T}_T^g, \boldsymbol{T}_T^o\}$
14: **Client executes**$(k, \{\boldsymbol{T}_t^g, \boldsymbol{T}_t^o\})$**:**
15: Assign prompts from server to local model $\{\boldsymbol{T}_{k,t}^g, \boldsymbol{T}_{k,t}^o\} \leftarrow \{\boldsymbol{T}_t^g, \boldsymbol{T}_t^o\}$
16: **for** each local epoch $e = 1, 2, ..., E$ **do**
17:     **for** batch of samples $(\boldsymbol{x}_{1:B}^k, \boldsymbol{y}_{1:B}^k) \in \mathcal{D}_k$ **do**
18:         Obtain the latent viusal representation for image data $\boldsymbol{h}_{1:B}^k = \mathcal{I}_{\boldsymbol{\theta}}(\boldsymbol{x}_{1:B}^k)$,
19:         Obtain the latent textual representations for prompts $\boldsymbol{e}_c = \mathcal{T}_{\boldsymbol{\theta}}(\boldsymbol{t}_c, \boldsymbol{n}_c)$ with $\boldsymbol{t}_c = (1-\rho)\boldsymbol{t}_c^l + \rho \boldsymbol{t}_c^g$, $\widetilde{\boldsymbol{e}}_u = \mathcal{T}_{\boldsymbol{\theta}}(\boldsymbol{t}_u^o, \boldsymbol{n}_u)$
20:         Compute the similarity scores $S(\boldsymbol{h}, \boldsymbol{e})$ between visual representations $\boldsymbol{h}_{1:B}^k$ and textual representations $\{\boldsymbol{e}_c\}_{c=1}^C$ and $\{\widetilde{\boldsymbol{e}}_u\}_{u=1}^U$
21:         Optimize prompts to build bi-level OOD separations by Algorithm 2
22:     **end for**
23: **end for**
24: **return** $\boldsymbol{\theta}_k^E$ to server

---

## C. Theoretical Analysis

**Definition C.1** (Optimal Transport Distance). The optimal transport distance is the distribution divergence between two probability masses $\hat{P}$ and $P_0$, i.e.,

$$D_{\text{OT}}(\hat{P}, P_0) = \inf_{\pi \in \boldsymbol{\pi}(\hat{P}, P_0), \hat{\boldsymbol{t}} \sim \hat{P}, \boldsymbol{t} \sim P_0} \int \mathfrak{c}(\hat{\boldsymbol{t}}, \boldsymbol{t}) d\pi \tag{12}$$
$$s.t. \ \boldsymbol{\pi}_1 = \boldsymbol{a}, \quad \boldsymbol{\pi}_2 = \boldsymbol{b},$$

with $\boldsymbol{\pi}(\hat{P}, P_0)$ is the couplings between $\hat{P}$ and $P_0$, $\boldsymbol{\pi}_1$ and $\boldsymbol{\pi}_2$ are marginals of $\boldsymbol{\pi}$ with assumption $\boldsymbol{a}^\top \mathbf{1} = \boldsymbol{b}^\top \mathbf{1}$, and $\mathfrak{c}(\hat{\boldsymbol{t}}, \boldsymbol{t})$ is short for the non-negative metric cost of samples $\hat{\boldsymbol{t}} \sim \hat{P}$ and $\boldsymbol{t} \sim P_0$.

**Definition C.2** (Unbalanced Optimal Transport Distance). The optimal transport distance is the distribution divergence

---

**Algorithm 2** Training procedure of Bi-level Distributional Robustness Optimization

---

**Input**: Batch size $B$, local steps $E$, dataset $\mathcal{D} = \cup_{k \in [K]} \mathcal{D}_k$

**Output**: Robust model parameters $\boldsymbol{\theta}$, Worst case global prompts and OOD prompts, i.e., $\boldsymbol{T}^g$ and $\boldsymbol{T}^o$

1: Sample $\boldsymbol{x}_{1:B} \sim D_k$
2: Initialize random noise for global prompts $\boldsymbol{\epsilon}_c^g \sim \mathcal{N}(0, \sigma I), \forall c \in \{1, \dots, C\}$
3: Perturb global prompts $\hat{\boldsymbol{t}}_c^g = \boldsymbol{t}_c^g + \boldsymbol{\epsilon}_c^g$
4: **for** Exploration step $n = 1$ to $N$ **do**
5: $\quad \boldsymbol{g}_{\boldsymbol{\epsilon}_c^g} = \nabla_{\boldsymbol{\epsilon}_c^g} \left[ \arg\min_{\hat{\boldsymbol{t}}^g} \mathbb{E}_{\boldsymbol{t}^g \sim P_0} \left[ \sup_{\hat{\boldsymbol{t}}^g \sim \hat{P}} \left\{ L(\hat{\boldsymbol{t}}^g) - \tau_1 \mathfrak{c}(\hat{\boldsymbol{t}}^g, \boldsymbol{t}^g) \right\} \right] - \gamma \|\boldsymbol{\epsilon}_c^g\|_1 \right]$
6: $\quad \boldsymbol{\epsilon}_c^g \leftarrow \boldsymbol{\epsilon}_c^g + lr \boldsymbol{g}_{\boldsymbol{\epsilon}_c^g}, \forall c \in \{1, \dots, C\}$
7: **end for**
8: Estimate robust global prompts $\hat{\boldsymbol{t}}^g$ via $\hat{\boldsymbol{t}}_c^g \leftarrow \boldsymbol{t}_c^g + \boldsymbol{\epsilon}_c^g$
9: Initialize random noise for global prompts $\boldsymbol{\epsilon}_u^o \sim \mathcal{N}(0, \sigma I), \forall u \in \{1, \dots, U\}$
10: Perturb OOD prompts $\hat{\boldsymbol{t}}_u^o = \boldsymbol{t}_u^o + \boldsymbol{\epsilon}_u^o$
11: **for** Exploration step $m = 1$ to $M$ **do**
12: $\quad \boldsymbol{o}_{\boldsymbol{\epsilon}_u^o} = \nabla_{\boldsymbol{\epsilon}_u^o} \left[ \arg\min_{\hat{\boldsymbol{t}}^o} \mathbb{E}_{\boldsymbol{t}^o \sim Q_0} \left[ \sup_{\hat{\boldsymbol{t}}^o \sim \hat{Q}} \left\{ L(\hat{\boldsymbol{t}}^o) - \tau_1 \mathfrak{c}(\hat{\boldsymbol{t}}^o, \boldsymbol{t}^o) \right\} \right] - \gamma \|\boldsymbol{\epsilon}_u^o\|_1 \right]$
13: $\quad \boldsymbol{\epsilon}_u^o \leftarrow \boldsymbol{\epsilon}_u^o + lr \boldsymbol{o}_{\boldsymbol{\epsilon}_u^o}, \forall u \in \{1, \dots, U\}$
14: **end for**
15: Estimate robust OOD prompts $\hat{\boldsymbol{t}}_u^o \leftarrow \boldsymbol{t}_u^o + \boldsymbol{\epsilon}_u^o$
16: Update parameters of $\boldsymbol{\theta}^t$ by computing the gradient of Eq. (30)

---

**Algorithm 3** Training procedure of Semi-unbalanced optimal transport based prompt calibration

---

**Input**: prompt sets $\{\boldsymbol{T}_{k,t}^g, \boldsymbol{T}_{k,t}^o\} \leftarrow$ from participating client $k$

**Output**: Robust model parameters $\boldsymbol{\theta}$, global-view consistent global prompts and OOD prompts, i.e., $\boldsymbol{T}^g$ and $\boldsymbol{T}^o$

1: Concatenate OOD prompts from clients $\boldsymbol{T}_{KU}^o \leftarrow \{\boldsymbol{T}_k^o\}_{k=1}^K$
2: Aggregate global prompts from clients by class $\boldsymbol{t}_c^g = \sum_{k=1}^K \frac{\mathcal{D}_{k,c}}{\sum_k \mathcal{D}_{k,c}} \boldsymbol{t}_{k,c}^g \, \forall c \in [C]$ and obtain $\boldsymbol{T}^g$ in server
3: Seek the semi-unbalanced optimal transport $\boldsymbol{\pi}^*$ between OOD prompts $\boldsymbol{T}_{KU}^o$ and $\boldsymbol{T}_s^g$
4: Estimate robust global prompts $\hat{\boldsymbol{t}}^g$ via Eq. (8)
5: Compute the matching probabilities $\boldsymbol{\pi}^{*\top} \mathbf{1}_{KU}$ to figure out top-M as seemly-OOD prompts
6: Update global prompts $\boldsymbol{T}_{t+1}^g$ with seemly-OOD prompts by Eq. (10)
7: Update OOD prompts $\boldsymbol{T}_{t+1}^o$ by Eq. (11) where top-U prompts satisfying negative matching probabilities $-\boldsymbol{\pi}^{*\top} \mathbf{1}_{KU}$

---

between two probability masses $\hat{Q}$ and $Q_0$, i.e.,

$$D_{\text{UOT}}(\hat{Q}, Q_0) = \inf_{\gamma \in \boldsymbol{\gamma}(\hat{Q}, Q_0), \hat{\boldsymbol{t}} \sim \hat{Q}, \boldsymbol{t} \sim Q_0} \int \mathfrak{c}(\hat{\boldsymbol{t}}, \boldsymbol{t}) d\gamma + \mu_1 D_{KL}(\boldsymbol{\gamma}_1 \| \hat{Q}) + \mu_2 D_{KL}(\boldsymbol{\gamma}_2 \| Q_0), \tag{13}$$

where $\boldsymbol{\gamma}(\hat{Q}, Q_0)$ is the couplings between $\hat{Q}$ and $Q_0$, $\boldsymbol{\gamma}_1$ and $\boldsymbol{\gamma}_2$ are marginals of $\boldsymbol{\gamma}$, $\mu_1$ and $\mu_2$ are regularization coefficient, $\mathfrak{c}(\hat{\boldsymbol{t}}, \boldsymbol{t})$ is short for the non-negative metric cost of samples $\hat{\boldsymbol{t}} \sim \hat{Q}$ and $\boldsymbol{t} \sim Q_0$. In terms of (Wang et al., 2025b), considering $\mu_1 = 0$, Eq. (13) can be rewritten as a special case:

$$\inf_{\hat{Q}, \gamma \in \boldsymbol{\gamma}(\hat{Q}, Q_0), \hat{\boldsymbol{t}} \sim \hat{Q}, \boldsymbol{t} \sim Q_0} \left\{ \int \mathfrak{c}(\hat{\boldsymbol{t}}, \boldsymbol{t}) d\gamma + \mu D_{KL}(\hat{Q} \| Q_0) \right\}. \tag{14}$$

**Theorem C.3.** *Suppose that the optimal dual variable $\tau_1^*$ and $\tau_2^*$ are strictly positive, bi-level separation loss $\mathcal{L}(\boldsymbol{x}, \boldsymbol{\theta}, \boldsymbol{t}^l, \boldsymbol{t}^g, \boldsymbol{t}^o)$ in Eq. (5) is concave and differentiable, Bi-level distribution robust optimization in Eq. (6) can be*

*solved via:*

$$
\begin{cases}
\hat{\boldsymbol{t}}^g = \arg\min_{\hat{\boldsymbol{t}}^g} \mathbb{E}_{\boldsymbol{t}^g \sim P_0} \left[ \sup_{\hat{\boldsymbol{t}}^g \sim \hat{P}} \left\{ L(\hat{\boldsymbol{t}}^g) - \tau_1 \mathfrak{c}(\hat{\boldsymbol{t}}^g, \boldsymbol{t}^g) \right\} \right], \\[2mm]
\hat{\boldsymbol{t}}^o = \arg\min_{\hat{\boldsymbol{t}}^o} \mathbb{E}_{\boldsymbol{t}^o \sim Q_0} \left[ \sup_{\hat{\boldsymbol{t}}^o \sim \hat{Q}} \left\{ L(\hat{\boldsymbol{t}}^o) - \tau_2 \mathfrak{c}\left(\hat{\boldsymbol{t}}^o, \boldsymbol{t}^o\right) \right\} \right]. \\[2mm]
\widehat{\mathcal{L}}^{BDRO} = \tau_2 \mu \log \mathbb{E}_{\hat{\boldsymbol{t}}^g, \hat{\boldsymbol{t}}^o} \left[ \exp(\dfrac{\mathcal{L}(\boldsymbol{x}, \boldsymbol{\theta}, \boldsymbol{t}^l, \hat{\boldsymbol{t}}^g, \hat{\boldsymbol{t}}^o) - \tau_1 \mathfrak{c}(\hat{\boldsymbol{t}}^g, \boldsymbol{t}^g) - \tau_2 \mathfrak{c}\left(\hat{\boldsymbol{t}}^o, \boldsymbol{t}^o\right)}{\tau_2 \mu}) \right],
\end{cases}
\tag{15}
$$

*where $\mu$ is the regularization of KL divergence, $L(\boldsymbol{t}^{\{\cdot\}})$ is short for $\mathcal{L}(\boldsymbol{x}, \boldsymbol{\theta}, \boldsymbol{t}^l, \boldsymbol{t}^g, \boldsymbol{t}^o)$ optimizing $\boldsymbol{t}^{\{\cdot\}}$.*

*Proof.* The OOD loss can be written as

$$
\begin{aligned}
\mathcal{L} &= \frac{1}{B} \sum_{b=1}^{B} [\mathcal{L}(\boldsymbol{x}_b, \boldsymbol{\theta}, \boldsymbol{t}^l, \boldsymbol{t}^g, \boldsymbol{t}^o)] \\
&= \frac{1}{B} \sum_{b=1}^{B} \mathbb{E}_{\boldsymbol{x} \in \mathcal{D}} \left[ -\log p(y = c | \boldsymbol{x}) p(y_{\text{ID}} = 1 | \boldsymbol{x}) \right],
\end{aligned}
\tag{16}
$$

with $p(y = c | \boldsymbol{x}) = \frac{S(\boldsymbol{x}, \boldsymbol{t}_c)}{\sum_{c=1}^{C} S(\boldsymbol{x}, \boldsymbol{t}_c) + \sum_{u=1}^{U} S(\boldsymbol{x}, \boldsymbol{t}_u^o)}$ and $p(y_{\text{ID}} = 1 | \boldsymbol{x}) = \frac{\sum_{c=1}^{C} S(\boldsymbol{x}, \boldsymbol{t}_c)}{\sum_{c=1}^{C} S(\boldsymbol{x}, \boldsymbol{t}_c) + \sum_{u=1}^{U} S(\boldsymbol{x}, \boldsymbol{t}_u^o)} = 1 - \frac{\sum_{u=1}^{U} S(\boldsymbol{x}, \boldsymbol{t}_u^o)}{\sum_{c=1}^{C} S(\boldsymbol{x}, \boldsymbol{t}_c) + \sum_{u=1}^{U} S(\boldsymbol{x}, \boldsymbol{t}_u^o)}$.

The corresponding distribution robust optimization objective is rewritten as:

$$
\begin{aligned}
\mathcal{L}_{BDRO}(\boldsymbol{x}, \boldsymbol{t}^l, \boldsymbol{t}^g, \boldsymbol{t}^o) &= \inf_{\boldsymbol{\theta}} \sup_{P \in \mathbb{P}, Q \in \mathbb{Q}} \mathbb{E}_{\hat{\boldsymbol{t}}^g \sim P, \hat{\boldsymbol{t}}^o \sim Q} \mathcal{L}(\boldsymbol{x}_b, \boldsymbol{\theta}, \boldsymbol{t}^l, \hat{\boldsymbol{t}}^g, \hat{\boldsymbol{t}}^o), \\
s.t. \quad &\begin{cases} \mathbb{P} = \{P \in \mathbb{D} : D_{\text{OT}}(P, P_0) \leq \eta_1\}, \\ \mathbb{Q} = \{Q \in \mathbb{D} : D_{\text{UOT}}(Q, Q_0) \leq \eta_2\}, \end{cases}
\end{aligned}
\tag{17}
$$

where $D_{\text{OT}}$ and $D_{\text{UOT}}$ are optimal transport distance and unbalanced optimal transport distance defined in Appendix C.2 and C.1.

First, we expand Eq. (17) with Lagrangian multipliers:

$$
\begin{aligned}
\mathcal{L}_{BDRO} &= \inf_{\tau_1 \geq 0, \tau_2 \geq 0} \sup_{\hat{P} \in \mathbb{P}, \hat{Q} \in \mathbb{Q}} \mathbb{E}_{\hat{\boldsymbol{t}}^g \sim P, \hat{\boldsymbol{t}}^o \sim Q} \left[ \mathcal{L}(\boldsymbol{x}_b, \boldsymbol{\theta}, \boldsymbol{t}^l, \hat{\boldsymbol{t}}^g, \hat{\boldsymbol{t}}^o) \right] - \tau_1 \left( D_{\text{OT}}(\hat{P}, P_0) - \eta_1 \right) - \tau_2 \left( D_{\text{UOT}}(\hat{Q}, Q_0) - \eta_2 \right) \\
&= \inf_{\tau_1 \geq 0, \tau_2 \geq 0} \sup_{\hat{P} \in \mathbb{P}, \hat{Q} \in \mathbb{Q}} \mathbb{E}_{\hat{\boldsymbol{t}}^g \sim \hat{P}, \hat{\boldsymbol{t}}^o \sim \hat{Q}} \left[ \mathcal{L}(\boldsymbol{x}_b, \boldsymbol{\theta}, \boldsymbol{t}^l, \hat{\boldsymbol{t}}^g, \hat{\boldsymbol{t}}^o) \right] - \tau_1 \left[ \inf_{\pi \in \boldsymbol{\pi}(P, P_0)} \int \mathfrak{c}(\hat{\boldsymbol{t}}^g, \boldsymbol{t}^g) d\pi(\hat{\boldsymbol{t}}^g, \boldsymbol{t}^g) - \eta_1 \right] \\
&\quad - \tau_2 \left[ \inf_{\gamma \in \boldsymbol{\gamma}(Q, Q_0)} \left( \int \mathfrak{c}(\hat{\boldsymbol{t}}^o, \boldsymbol{t}^o) d\gamma(\hat{\boldsymbol{t}}^o, \boldsymbol{t}^o) + \mu D_{KL}(\hat{Q} \| Q_0) \right) - \eta_2 \right].
\end{aligned}
\tag{18}
$$

Then we optimize for the worst-case distribution of global prompts $\boldsymbol{t}^g$ that are independent with ood prompts $\boldsymbol{t}^o$, meaning that we can treat $\hat{Q}$ as constant. By denoting $\mathcal{L}(\boldsymbol{x}_b, \boldsymbol{\theta}, \boldsymbol{t}^l, \hat{\boldsymbol{t}}^g, \hat{\boldsymbol{t}}^o)$ as $L(\hat{\boldsymbol{t}}^g)$, if $L(\hat{\boldsymbol{t}}^g)$ is upper semi-continuous, we can obtain the dual form of Eq. (17) with regarding to $\hat{P}$ (Bui et al., 2022; Sinha et al., 2017), which can be formulated as:

$$
\begin{aligned}
&\inf_{\tau_1 \geq 0} \sup_{\hat{P} \in \mathbb{P}} \mathbb{E}_{\hat{\boldsymbol{t}}^g \sim \hat{P}} \left[ L(\hat{\boldsymbol{t}}^g) \right] - \tau_1 \left[ D_{\text{OT}}(\hat{P}, P_0) - \eta_1 \right] \\
&= \inf_{\tau_1 \geq 0} \sup_{\hat{P} \in \mathbb{P}} \mathbb{E}_{\hat{\boldsymbol{t}}^g \sim \hat{P}} \left[ L(\hat{\boldsymbol{t}}^g) \right] - \tau_1 \left[ \mathbb{E}_{\hat{\boldsymbol{t}}^g \sim \hat{P}} \mathbb{E}_{\boldsymbol{t}^g \sim P_0} \mathfrak{c}(\hat{\boldsymbol{t}}^g, \boldsymbol{t}^g) - \eta_1 \right] \\
&= \inf_{\tau_1 \geq 0} \left\{ \tau_1 \eta_1 + \mathbb{E}_{\boldsymbol{t}^g \sim P_0} \left[ \sup_{\hat{P} \in \mathbb{P}} \mathbb{E}_{\hat{\boldsymbol{t}}^g \sim \hat{P}} \left\{ L(\hat{\boldsymbol{t}}^g) - \tau_1 \mathfrak{c}(\hat{\boldsymbol{t}}^g, \boldsymbol{t}^g) \right\} \right] \right\} \\
&= \mathbb{E}_{\boldsymbol{t}^g \sim P_0} \left[ \sup_{\hat{\boldsymbol{t}}^g \sim \hat{P}} \left\{ L(\hat{\boldsymbol{t}}^g) - \tau_1 \mathfrak{c}(\hat{\boldsymbol{t}}^g, \boldsymbol{t}^g) \right\} \right],
\end{aligned}
\tag{19}
$$

where the last equation holds by relaxing $\tau_1 \geq 0$ (Sinha et al., 2017). Thus we can obtain the worst case of global prompts via

$$\hat{\boldsymbol{t}}^g = \arg\min_{\hat{\boldsymbol{t}}^g} \mathbb{E}_{\boldsymbol{t}^g \sim P_0} \left[ \sup_{\hat{\boldsymbol{t}}^g \sim \hat{P}} \left\{ L(\hat{\boldsymbol{t}}^g) - \tau_1 \mathfrak{c}(\hat{\boldsymbol{t}}^g, \boldsymbol{t}^g) \right\} \right]. \tag{20}$$

Similarly, we can fix $\hat{P}$ to optimize $\hat{Q}$ by denoting $\mathcal{L}(\boldsymbol{x}_b, \boldsymbol{\theta}, \hat{\boldsymbol{t}}^l, \hat{\boldsymbol{t}}^g, \hat{\boldsymbol{t}}^o)$ as $L(\hat{\boldsymbol{t}}^o)$, as below:

$$
\begin{aligned}
&\inf_{\tau_2 \geq 0} \sup_{\hat{Q} \in \mathbb{Q}} \mathbb{E}_{\hat{\boldsymbol{t}}^o \sim \hat{Q}}[L(\hat{\boldsymbol{t}}^o)] - \tau_2 \left[ D_{\text{UOT}}(\hat{Q}, Q_0) - \eta_2 \right] \\
&= \inf_{\tau_2 \geq 0} \tau_2 \eta_2 + \sup_{\hat{Q} \in \mathbb{Q}} \left\{ \mathbb{E}_{\hat{\boldsymbol{t}}^o \sim \hat{Q}}[L(\hat{\boldsymbol{t}}^o)] - \tau_2 \mathbb{E}_{(\hat{\boldsymbol{t}}^o, \boldsymbol{t}^o) \sim \gamma} \left[ c\left(\hat{\boldsymbol{t}}^o, \boldsymbol{t}^o\right) + \mu D_{KL}\left(\hat{Q} \| Q_0\right) \right] \right\} \\
&= \inf_{\tau_2 \geq 0} \tau_2 + \sup_{\hat{Q} \in \mathbb{Q}, \boldsymbol{\gamma}(\hat{P}, P_0)} \mathbb{E}_{\hat{\boldsymbol{t}}^o \sim \hat{Q}} \mathbb{E}_{(\hat{\boldsymbol{t}}^o, \boldsymbol{t}^o) \sim \gamma} \left[ L(\hat{\boldsymbol{t}}^o) - \tau_2 c\left(\hat{\boldsymbol{t}}^o, \boldsymbol{t}^o\right) - \tau_2 \mu \log \frac{\hat{Q}\left(\hat{\boldsymbol{t}}^o\right)}{Q_0\left(\hat{\boldsymbol{t}}^o\right)} \right] \\
&= \inf_{\tau_2 \geq 0} \tau_2 + \sup_{\hat{Q} \in \mathbb{Q}} \mathbb{E}_{\hat{\boldsymbol{t}}^o \sim \hat{Q}} \left[ \sup_{\hat{\boldsymbol{t}}^o \sim \hat{Q}} \left\{ L(\hat{\boldsymbol{t}}^o) - \tau_2 c\left(\hat{\boldsymbol{t}}^o, \boldsymbol{t}^o\right) \right\} - \tau_2 \mu \log \frac{\hat{Q}\left(\hat{\boldsymbol{t}}^o\right)}{Q_0\left(\hat{\boldsymbol{t}}^o\right)} \right]
\end{aligned}
\tag{21}
$$

Let $\mathfrak{Q} = \frac{\hat{Q}(\hat{\boldsymbol{t}}^o)}{Q_0(\hat{\boldsymbol{t}}^o)}$ with the probability constraint $\mathbb{E}[\mathfrak{Q}] = 1$, $f(\hat{\boldsymbol{t}}^o) = \sup_{\hat{\boldsymbol{t}}^o \sim \hat{Q}} \left\{ L(\hat{\boldsymbol{t}}^o) - \tau_2 c\left(\hat{\boldsymbol{t}}^o, \boldsymbol{t}^o\right) \right\}$, the sencond term of Eq. (21) can be rewritten as:

$$
\begin{aligned}
\mathcal{J} &= \mathbb{E}_{\hat{\boldsymbol{t}}^o \sim \hat{Q}} \left[ \sup_{\hat{\boldsymbol{t}}^o \sim \hat{Q}} \left\{ L(\hat{\boldsymbol{t}}^o) - \tau_2 c\left(\hat{\boldsymbol{t}}^o, \boldsymbol{t}^o\right) \right\} - \tau_2 \mu \log \frac{\hat{Q}\left(\hat{\boldsymbol{t}}^o\right)}{Q_0\left(\hat{\boldsymbol{t}}^o\right)} \right] \\
&= \mathbb{E}_{\hat{\boldsymbol{t}}^o \sim Q_0} \left[ \frac{\hat{Q}\left(\hat{\boldsymbol{t}}^o\right)}{Q_0\left(\hat{\boldsymbol{t}}^o\right)} \sup_{\hat{\boldsymbol{t}}^o \sim \hat{Q}} \left\{ L(\hat{\boldsymbol{t}}^o) - \tau_2 c\left(\hat{\boldsymbol{t}}^o, \boldsymbol{t}^o\right) \right\} - \tau_2 \mu \frac{\hat{Q}\left(\hat{\boldsymbol{t}}^o\right)}{Q_0\left(\hat{\boldsymbol{t}}^o\right)} \log \frac{\hat{Q}\left(\hat{\boldsymbol{t}}^o\right)}{Q_0\left(\hat{\boldsymbol{t}}^o\right)} \right] \\
&= \begin{cases} \mathbb{E}_{\boldsymbol{t}^o \sim Q_0} \left[ \mathfrak{Q} f(\hat{\boldsymbol{t}}^o) - \tau_2 \mu \mathfrak{Q} log \mathfrak{Q} \right] \\ s.t. \mathbb{E}[\mathfrak{Q}] = 1 \end{cases}
\end{aligned}
\tag{22}
$$

The Lagrangian expansion is

$$\mathcal{J} = \inf_{\mu \geq 0, \nu \geq 0, \tau_2 \geq 0} \mathbb{E}_{\boldsymbol{t}^o \sim Q_0} \left[ \mathfrak{Q} f(\hat{\boldsymbol{t}}^o) - \tau_2 \mu \mathfrak{Q} log \mathfrak{Q} \right] - \nu \left( \mathbb{E}[\mathfrak{Q}] - 1 \right), \tag{23}$$

where $\nu$ is the multiplier. Dy deviating,

$$\frac{\partial \mathcal{J}}{\partial \mathfrak{Q}} = f(\hat{\boldsymbol{t}}^o) - \tau_2 \mu (\log \mathfrak{Q} + 1) - \nu = 0. \tag{24}$$

The optimal $\mathfrak{Q}^*$ is realized

$$\mathfrak{Q}^* = \exp(\frac{f(\hat{\boldsymbol{t}}^o) - \nu}{\tau_2 \mu} - 1). \tag{25}$$

Then we take the optimal $\mathfrak{Q}^*$ back to Eq. (23), we can achieve:

$$
\begin{aligned}
\mathcal{J} &= \inf_{\mu \geq 0, \nu \geq 0, \tau_2 \geq 0} \tau_2 \mu \mathbb{E}_{\hat{\boldsymbol{t}} \sim P_0} \left[ \exp(\frac{f(\hat{\boldsymbol{t}}^o) - \nu}{\tau_2 \mu} - 1) \right] + \nu \\
&= \tau_2 \mu \exp(-\frac{\nu}{\tau_2 \mu} - 1) \mathbb{E}_{\hat{\boldsymbol{t}} \sim P_0} \left[ \exp(\frac{f(\hat{\boldsymbol{t}}^o)}{\tau_2 \mu}) \right] + \nu.
\end{aligned}
\tag{26}
$$

Since $\nu^*$ holds when its gradient equals to 0, i.e.,

$$\frac{\partial \mathcal{J}}{\partial \nu} = -\exp\left(-\frac{\nu}{\tau_2 \mu} - 1\right) \mathbb{E}_{\boldsymbol{x} \sim P(x)} \left[ \exp\left(\frac{f(\hat{\boldsymbol{t}}^o)}{\tau_2 \mu}\right) \right] + 1 = 0. \tag{27}$$

We have solution $\nu^* = \tau_2\mu \log \mathbb{E}_{\boldsymbol{x}\sim P(\boldsymbol{x})}\left[\exp\left(\frac{f(\hat{\boldsymbol{t}}^o)}{\tau_2\mu}\right)\right] - \tau_2\mu$ and finally:

$$\min_{\tau_2\mu\geq 0} \mathcal{J} = \max_{\hat{\boldsymbol{t}}^o\sim\hat{Q}} \tau_2\mu \log \mathbb{E}_{\boldsymbol{t}^o\sim P_0(\boldsymbol{t})}\left[\exp\left(\frac{f(\hat{\boldsymbol{t}}^o)}{\tau_2\mu}\right)\right]. \tag{28}$$

Finally, it achieves supremum of $\hat{Q} \in \mathbb{P}$:

$$\inf_{\tau_2\geq 0} \sup_{\hat{Q}\in\mathbb{Q}} \mathbb{E}_{\hat{\boldsymbol{t}}^o\sim\hat{Q}}[L(\hat{\boldsymbol{t}}^o)] - \tau_2\left[D_{\text{UOT}}(\hat{Q},Q_0) - \eta_2\right]$$

$$= \inf_{\tau_2\geq 0} \tau_2\eta_2 + \sup_{\hat{Q}\in\mathbb{Q}} \tau_2\mu \log \mathbb{E}_{\boldsymbol{t}^o\sim P_0}\left[\exp(\frac{f(\hat{\boldsymbol{t}}^o)}{\tau_2\mu})\right]. \tag{29}$$

Remind that the model parameters are independent with $\tau_1$, $\tau_2$ and $\mu$, we can treat them as hyperparameters with positive values in optimizing Eq. (17), and finally update model with prompts by the gradient of loss

$$\widehat{\mathcal{L}}^{\text{BDRO}} = \tau_2\mu \log \mathbb{E}_{\hat{\boldsymbol{t}}^g,\hat{\boldsymbol{t}}^o}\left[\exp(\frac{\mathcal{L}(\boldsymbol{x},\boldsymbol{\theta},\boldsymbol{t}^l,\hat{\boldsymbol{t}}^g,\hat{\boldsymbol{t}}^o) - \tau_1\mathfrak{c}(\hat{\boldsymbol{t}}^g,\boldsymbol{t}^g) - \tau_2\mathfrak{c}\left(\hat{\boldsymbol{t}}^o,\boldsymbol{t}^o\right)}{\tau_2\mu})\right]. \tag{30}$$

Similarly, we can obtain the worst case OOD prompts via

$$\hat{\boldsymbol{t}}^o = \arg\min_{\hat{\boldsymbol{t}}^o} \mathbb{E}_{\boldsymbol{t}^o\sim Q_0}\left[\sup_{\hat{\boldsymbol{t}}^o\sim\hat{Q}}\left\{L(\hat{\boldsymbol{t}}^o) - \tau_2\mathfrak{c}\left(\hat{\boldsymbol{t}}^o,\boldsymbol{t}^o\right)\right\}\right]. \tag{31}$$

Worth to mention that, the values of $\eta_1$ and $\eta_2$ are constants in the optimization, which is directly controlled via $\tau_1$ and $\tau_2$, respectively. $\square$

**Theorem C.4.** *Semi-Unbalanced Optimal Transport Optimization of Eq.* (8) *can be solved by Frank-Wolfe Algorithm (Clarkson, 2010; Jaggi, 2013), which iteratively updates* $\boldsymbol{\pi}^{i+1} = \boldsymbol{\pi}^i - \beta(\boldsymbol{\pi}^i - \boldsymbol{s}^i)$, *with step size* $\beta = \frac{1}{i+2}$ *following Armijo condition (Armijo, 1966) and optimal directions* $\boldsymbol{s}^i$ *satisfying*

$$s_{cj}^i = \begin{cases} b_j^i, & \text{if} \quad c^i = \arg\min_c \nabla J_{\text{SemiUOT}}\left(\boldsymbol{\pi}^i_{\cdot j}\right) \\ 0, & \text{otherwise}. \end{cases} \tag{32}$$

*with initialization* $s_{cj}^0 = \begin{cases} b_j^0, & c=0 \\ 0, & \text{otherwise}. \end{cases}$

*Proof.* Now we expand semi-unbalanced optimal transport as below:

$$\min_{\boldsymbol{\pi}\geq 0} J_{\text{SemiUOT}} = \begin{cases} \langle \mathfrak{C},\boldsymbol{\pi}\rangle + \lambda\text{KL}(\boldsymbol{\pi}^\top\mathbf{1}_{KU}\|\boldsymbol{a}) \\ \text{s.t.} \quad \boldsymbol{\pi}\mathbf{1}_C = \boldsymbol{b}, \pi_{cj}\geq 0 \quad \forall_{j\in[KU]}, \end{cases}$$

$$= \begin{cases} \sum_{c=1}^C\sum_{j=1}^{KU}\pi_{cj}\mathfrak{C}_{cj} + \lambda\sum_{c=1}^C\left[\sum_{j=1}^{KU}\pi_{cj}\log\frac{\sum_{m=1}^{KU}\pi_{cj}}{a_c} - \sum_{m=1}^{KU}\pi_{cj} + a_c\right] \\ \text{s.t.} \sum_{c=1}^C\pi_{cj} = b_j, \quad \pi_{cj}\geq 0 \quad \forall_{j\in[KU]}. \end{cases} \tag{33}$$

Then we can optimize the problem via Frank-Wolfe algorithm (Clarkson, 2010; Jaggi, 2013), i.e.,

$$\arg\min_{\boldsymbol{s}^i} \ell = \sum_{c=1}^C\sum_{j=1}^{KU}s_{cj}^i\nabla J\left(\pi_{cj}^i\right) = \sum_{c=1}^C\sum_{j=1}^{KU}s_{cj}^i\left[\mathfrak{C}_{cj} + \tau\log\frac{\sum_{m=1}^{KU}\pi_{cm}^i}{a_c}\right]$$

$$\text{s.t.} \sum_{c=1}^C\pi_{cj}^i = b_j, \quad \pi_{cj}^i\geq 0. \tag{34}$$

The solution can be assigned, i.e.,

$$s_{cj}^i = \arg\min_{s_{cj}^i} \sum_{i=1}^{M} \sum_{j=1}^{N} s_{cj}^i \left[ \mathfrak{C}_{cj} + \tau \log \frac{\sum_{m=1}^{KU} \pi_{cm}^i}{a_c} \right]. \tag{35}$$

Notably, Eq. (35) is a linear function, where the minimum holds on the minimal values of $\nabla J\left(\pi_{cj}^i\right)$. Since $\pi_{cj} \geq 0$, $\nabla J\left(\pi_{cj}^i\right) = \left[ \mathfrak{C}_{cj} + \tau \log \frac{\sum_{m=1}^{KU} \pi_{cm}^i}{a_c} \right]$ is the monotonically increasing function whose minimal value is determined via columne-wise, thus we can achieve:

$$s_{cj}^i = \begin{cases} b_j, & \text{if} \quad c^i = \arg\min_{c^i} \nabla J_{\text{SemiUOT}}\left(\boldsymbol{\pi}_{\cdot j}^i\right) \\ 0, & \text{otherwise}. \end{cases} \tag{36}$$

Then we can update $\boldsymbol{\pi}$ via

$$\boldsymbol{\pi}^{i+1} = (1-\beta)\boldsymbol{\pi}^i + \beta\boldsymbol{s}^i, \tag{37}$$

where $\widehat{\boldsymbol{\pi}}$ is the value of in last updating iteration and $\mu$ is the step-size which can be updated with Armijo linear search methods (Armijo, 1966), e.g., $\beta = 1/(i+2)$. In the beginning, we can assign $s_{cj}^0 = \begin{cases} b_j^0, & c = 0 \\ 0, & \text{otherwise} \end{cases}$ as our initial point, and $\boldsymbol{\pi}$ will penalize the wrong guess and adjust the optimal direction $\boldsymbol{s}$ iteratively. $\qquad \square$

## D. Datasets and Implementation Details

**Datasets.** We study the OOD robustness of federated prompt learning on fifteen datasets: CIFAR-100 (Krizhevsky et al., 2009) and TinyImageNet (Le & Yang, 2015) for both generalization and detection, Food101 (Bossard et al., 2014), DTD (Cimpoi et al., 2014a), Caltech101 (Fei-Fei et al., 2004), Flowers (Nilsback & Zisserman, 2008), and OxfordPet (Parkhi et al., 2012) for label shift generalization, DomainNet (Peng et al., 2019), Office-Caltech10 (Gong et al., 2012), and PACS (Li et al., 2017) for feature-shift (domain) generalization, CIFAR-100-C, TinyImageNet-C (Hendrycks & Dietterich, 2018) for covariate-shift generalization, as well as Places365 (Zhou et al., 2017), Texture (Cimpoi et al., 2014b), iSUN (Xu et al., 2015), LSUN-C and LSUN-R (Yu et al., 2015) for detection. We provide the summary of data in Tab. 6. **In terms of maintaining performance and OOD robustness**, we simulate heterogeneous distribution following both Dirichlet and Pathlogical settings (McMahan et al., 2017; Li et al., 2020) on CIFAR-100 (Krizhevsky et al., 2009) and TinyImageNet (Le & Yang, 2015) as conventional work does (Liao et al., 2024b). We test the generaliazation based on CIFAR-100-C (Hendrycks & Dietterich, 2018) and TinyImageNet-C (Le & Yang, 2015). Meanwhile, we study on iNaturalist (Van Horn et al., 2018), iSUN (Xiao et al., 2010), Place (Zhou et al., 2017), and Textures (Cimpoi et al., 2014b), following existing CLIP-based OOD detection methods (Wang et al., 2023; Miyai et al., 2024). **To widely evaluate OOD generalization and detection**, we follow previous work of federated prompt learning (Cui et al., 2024; Guo et al., 2023b;a), to study (1) heterogeneous label shift generalization on Food101 (Bossard et al., 2014), DTD (Cimpoi et al., 2014a), Caltech101 (Fei-Fei et al., 2004), Flowers (Nilsback & Zisserman, 2008), and OxfordPet (Parkhi et al., 2012) to predict the accuracy of personalization following pathological heterogeneity, and (2) feature shift domain generalization on DomainNet (Peng et al., 2019), and Office-Caltech10 (Gong et al., 2012), by leave-one-domain-out validation strategy (Nguyen et al., 2022b). Specifically, for $N-1$ domains of one dataset, we train each client with distinct domain data, and test its model generalization on the whole target data of remaining one domain.

**Comparison Methods.** We categorize the comparison methods into two types, i.e., (1) **Existing federated prompt learning methods for VLMs generalization**: pFedprompt (Guo et al., 2023a), PromptFL (Guo et al., 2023b), FedOTP (Li et al., 2024), FedPGP (Cui et al., 2024), PromptFolio (Pan et al., 2024), and (2) **Adapting existing centralized OOD robustness methods for federated scenarios**: FedGalLoP (Lafon et al., 2025), FedLoCoOp (Miyai et al., 2024), and FedLAPT (Zhang et al., 2025). We select baselines from the state-of-the-art (SOTA) personalized federated learning (PFL) methods and centralized prompt-based OOD detection methods. For centralized prompt-based OOD baselines, we choose GalLop and LAPT due to their strong performance in both generalization and OOD detection. We also include LoCoOp, as GalLop is proposed as an improvement over it.

- PromptFL (Guo et al., 2023b) replaces the federated model training with the federated prompt training, accelerating both the local training and the global aggregation.

| Dataset | Classes | Train | Test | Domains | Task |
|---|---|---|---|---|---|
| CIFAR100 (Krizhevsky et al., 2009) | 100 | 50,000 | 10,000 | 1 | Generalization and Detection |
| TinyImageNet (Le & Yang, 2015) | 200 | 100,000 | 10,000 | 1 | |
| Food101 (Bossard et al., 2014) | 101 | 50,500 | 30,300 | 1 | |
| DTD (Cimpoi et al., 2014a) | 47 | 2,820 | 1,692 | 1 | |
| Caltech101 (Fei-Fei et al., 2004) | 100 | 4,128 | 2,465 | 1 | Label Shift Generalization |
| Flowers (Nilsback & Zisserman, 2008) | 102 | 4.093 | 2,463 | 1 | |
| OxfordPet (Parkhi et al., 2012) | 37 | 2,944 | 3,669 | 1 | |
| DomainNet (Peng et al., 2019) | 10 | 18,278 | 4,573 | 6 | |
| Office-Caltech10 (Gong et al., 2012) | 10 | 2,025 | 508 | 4 | Feature Shift (Domain) Generalization |
| CIFAR-100-C (Hendrycks & Dietterich, 2018) | 100 | 50,000 | 10,000 | 1 | Covariate-Shift Generalization |
| TinyImageNet-C (Hendrycks & Dietterich, 2018) | 200 | 100,000 | 10,000 | 1 | |
| Places365 (Zhou et al., 2017) | 434 | 18,000,000 | 36,000 | 1 | |
| Texture (Cimpoi et al., 2014b) | 47 | 2,820 | 1,692 | 1 | |
| iSUN (Xu et al., 2015) | 813 | 50,000 | 12,000 | 1 | Detection |
| LSUN-C (Yu et al., 2015) | 10 | 50,000 | 10,000 | 1 | |
| LSUN-R (Yu et al., 2015) | 10 | 50,000 | 10,000 | 1 | |

Table 6: Statistical details of datasets used in experiments.

- FedOTP (Li et al., 2024) provides each client with a global prompt and a local prompt and utilizes unbalanced Optimal Transport to align local visual features with these prompts.

- FedPGP (Cui et al., 2024) uses low-rank decomposition to adapt global prompts to heterogeneous local distributions and integrate an extra contrastive loss, considering both personalization and generalization.

- PromptFolio (Pan et al., 2024) analyzes via feature learning theory and combines global and local prompts into a prompt portfolio to balance generalization and personalization.

- FedGalLoP is a federated version of GalLoP (Lafon et al., 2025). GalLoP learns multiple diverse prompts leveraging both global and local visual features, enforcing prompt diversity using the "prompt dropout" technique.

- FedLoCoOp is a federated version of LoCoOp (Miyai et al., 2024). LoCoOp uses local regularization to minimize ID-irrelevant nuisances in CLIP features, improving the separation between ID and OOD classes.

- FedLAPT is a federated version of LAPT (Zhang et al., 2025). LAPT reduces the need for manual prompt engineering by automatically generating distribution-aware prompts.

**Implementation Details and Evaluation Metrics.** We conduct experiments on ViT-B/16 (Dosovitskiy, 2020) CLIP models. To study the heterogeneity generalization on CIFAR-100/TinyImageNet datasets, we simulate both cross-device and cross-silo scenarios. That is, we set local training epoch $E = 2$, communication round $T = 25$, and the number of clients $K = 10$ for fully participation. While in cross-device setting, we choose local training epochs $E = 2$, communication rounds $T = 100$, and $K = 100$ for 10% participation. To obtain fair comparisons, all comparison methods are tuned for converging using their best hyper-parameters, and we report the average of the results from three random seeds. We set the learnable prompt vectors with length as 16, embedding size as 512, class token position as 'end', and random initialization. We choose 1 prompt per class for both local and global ID prompts, and 100 OOD prompts in total. We report the average Top-1 accuracies for generalization of ID (ACC↑) and ID-C (CACC↑). We compute maximum concept matching (MCM) (Ming et al., 2022) as OOD detection score, which is based on similarity between textual features and image features. Based on MCM, we report the standard metrics used for OOD detection, i.e., AUROC (↑) and FPR95 (↓) (Yang et al., 2024).

# E. Additional Experimental Results

In this section, we report the results on CIFAR-100 and TinyImageNet under different Dirichlet distributions (Tables 6–7), which are consistent with Tables 1–3 and further verify OOD robustness. Table 4 has been split into Tables 8 and 9 for

Table 7: Main results of federated prompt learning on CIFAR-100 with different Dirichlet distributions ($K = 10$).

| Methods | $\alpha = 0.1$ | | | | $\alpha = 0.5$ | | | | $\alpha = 5.0$ | | | |
| --- | --- | --- | --- | --- | --- | --- | --- | --- | --- | --- | --- | --- |
| | ACC | CACC | FPR95 | AUROC | ACC | CACC | FPR95 | AUROC | ACC | CACC | FPR95 | AUROC |
| PromptFL | 71.22 | 67.55 | 76.58 | 72.20 | 75.65 | 71.52 | 82.13 | 69.65 | 74.92 | 71.37 | 79.52 | 74.25 |
| FedOTP | 76.81 | 73.50 | 61.88 | 79.14 | 68.43 | 65.67 | 73.78 | 73.45 | 66.20 | 63.16 | 77.73 | 71.15 |
| FedPGP | 76.77 | 72.55 | 74.81 | 74.45 | 72.95 | 69.25 | 83.65 | 71.37 | 73.01 | 69.15 | 82.57 | 72.65 |
| PromptFolio | 80.07 | 76.89 | 65.30 | 77.95 | 75.98 | 71.98 | 78.61 | 71.44 | 74.19 | 70.60 | 79.64 | 72.74 |
| FedLoCoOp | 67.87 | 63.70 | 76.81 | 70.40 | 74.44 | 70.35 | 73.28 | 72.56 | 74.87 | 70.98 | 74.82 | 73.72 |
| FedGalLoP | 80.53 | 77.61 | 60.72 | 82.66 | 75.87 | 72.85 | 68.72 | 79.66 | 74.32 | 71.14 | 72.72 | 79.13 |
| FedLAPT | 61.20 | 57.54 | 80.28 | 69.97 | 59.41 | 56.33 | 81.97 | 66.73 | 60.03 | 56.29 | 80.13 | 68.42 |
| FOCoOp | **82.42** | **78.52** | **46.56** | **86.98** | **77.71** | **73.59** | **54.26** | **83.40** | **77.66** | **73.59** | **51.02** | **83.22** |
| -w/o-BOS | 79.18 | 76.04 | 54.30 | 82.34 | 74.39 | 70.55 | 58.40 | 81.27 | 75.09 | 71.76 | 54.92 | 81.64 |
| -w/o-GOC | 78.66 | 75.99 | 53.97 | 82.56 | 74.78 | 70.88 | 57.83 | 81.55 | 75.04 | 71.50 | 55.20 | 81.85 |

Table 8: Main results of federated prompt learning on TinyImageNet with different Dirichlet distributions ($K = 10$).

| Methods | $\alpha = 0.1$ | | | | $\alpha = 0.5$ | | | | $\alpha = 5.0$ | | | |
| --- | --- | --- | --- | --- | --- | --- | --- | --- | --- | --- | --- | --- |
| | ACC | CACC | FPR95 | AUROC | ACC | CACC | FPR95 | AUROC | ACC | CACC | FPR95 | AUROC |
| PromptFL | 70.29 | 63.41 | 69.38 | 73.09 | 73.22 | 65.57 | 68.91 | 75.39 | 73.09 | 66.27 | 71.04 | 74.32 |
| FedOTP | 70.36 | 64.49 | 71.82 | 69.75 | 63.32 | 57.87 | 78.70 | 63.59 | 60.94 | 55.44 | 81.14 | 62.17 |
| FedPGP | 74.10 | 67.45 | 66.65 | 75.01 | 71.97 | 64.77 | 68.68 | 73.83 | 70.92 | 64.82 | 70.73 | 74.00 |
| PromptFolio | 78.09 | 71.78 | 61.24 | 78.78 | 74.08 | 66.60 | 68.83 | 75.23 | 71.73 | 65.91 | 70.23 | 74.48 |
| FedLoCoOp | 65.97 | 58.72 | 70.47 | 72.24 | 71.92 | 63.64 | 67.97 | 75.24 | 72.51 | 64.13 | 65.78 | 75.13 |
| FedGalLoP | 79.08 | 73.00 | 58.37 | 80.60 | 74.95 | 68.70 | 64.20 | 79.01 | 72.63 | 66.91 | 65.48 | 77.96 |
| FedLAPT | 59.82 | 54.97 | 75.34 | 69.74 | 59.66 | 54.72 | 74.98 | 70.72 | 59.88 | 54.58 | 75.93 | 69.34 |
| FOCoOp | **81.58** | **74.74** | **45.44** | **85.16** | **76.31** | **70.06** | **48.92** | **84.42** | **74.41** | **68.80** | **49.86** | **83.17** |
| -w/o-BOS | 79.49 | 72.45 | 51.44 | 82.38 | 74.96 | 68.33 | 53.29 | 81.12 | 73.81 | 66.80 | 54.73 | 80.45 |
| -w/o-GOC | 78.63 | 71.59 | 54.22 | 80.99 | 73.53 | 66.76 | 55.37 | 80.48 | 72.98 | 66.13 | 56.33 | 79.92 |

clarity. Tables 10–11 provide numerical results for Figure 4 (OOD detection on various OUT datasets), while Tables 12–13 are numerical results corresponding to Figure 5 (generalization on different ID-C datasets). Figures 8–9 are also enlarged for better readability.

Table 9: Domain generalization on DomainNet.

| Method \Domain | Clipart | Infograph | Painting | Quickdraw | Real | Sketch | average |
|---|---|---|---|---|---|---|---|
| PromptFL | 96.28 | 74.84 | 95.81 | 60.28 | 96.77 | 96.36 | 86.72 |
| FedOTP | 91.03 | 61.52 | 86.98 | 53.04 | 91.16 | 89.73 | 78.91 |
| FedPGP | 93.67 | 75.07 | 93.62 | 58.09 | 95.44 | 95.48 | 85.23 |
| PromptFolio | 95.24 | 75.64 | 94.78 | 59.02 | 95.58 | 95.41 | 85.95 |
| FedLoCoOp | 95.34 | 72.31 | 92.78 | 60.11 | 96.07 | 96.12 | 85.46 |
| FedGalLoP | 95.62 | 75.40 | 94.78 | 65.08 | 96.23 | 96.71 | 87.30 |
| FedLAPT | 92.36 | 66.54 | 89.02 | 48.38 | 94.07 | 92.07 | 80.41 |
| FOCoOp | **96.44** | **76.59** | **96.72** | 62.99 | **97.16** | 96.22 | **87.68** |

Table 10: Domain generalization on Office.

| Method \Domain | Amazon | Caltech | DSLR | WebCam | Avg |
|---|---|---|---|---|---|
| PromptFL | 96.21 | 94.64 | 99.20 | 97.03 | 96.77 |
| FedOTP | 94.64 | 93.15 | 98.93 | 96.46 | 95.80 |
| FedPGP | 95.17 | 95.36 | 99.73 | 97.45 | 96.93 |
| PromptFolio | 96.73 | 94.29 | 98.66 | 97.59 | 96.82 |
| FedLoCoOp | 96.47 | 94.15 | 93.60 | 95.33 | 94.89 |
| FedGalLoP | 97.30 | 96.33 | 99.73 | 98.58 | 97.99 |
| FedLAPT | 77.28 | 84.61 | 86.40 | 86.86 | 83.79 |
| FOCoOp | **98.21** | **96.54** | 99.71 | 98.20 | **98.16** |

Table 11: Detection results on CIFAR-100 non-overlap pathological heterogeneity.

| Dataset | INaturalist | | Texture | | iSUN | | Places | |
|---|---|---|---|---|---|---|---|---|
| Method | FPR95 | AUROC | FPR95 | AUROC | FPR95 | AUROC | FPR95 | AUROC |
| PromptFL | 78.23 | 64.51 | 84.51 | 68.28 | 65.57 | 78.55 | 90.24 | 53.50 |
| FedOTP | 43.20 | 85.26 | 38.22 | 87.56 | 35.61 | 89.44 | 42.92 | 87.22 |
| FedPGP | 49.01 | 82.21 | 51.57 | 84.68 | 44.62 | 86.78 | 63.78 | 77.91 |
| PromptFolio | 40.23 | 87.02 | 44.26 | 88.06 | 35.39 | 89.58 | 54.57 | 82.73 |
| FedLoCoOp | 94.26 | 51.63 | 77.59 | 68.76 | 61.92 | 82.80 | 87.01 | 60.45 |
| FedGalLoP | 34.41 | 89.29 | 41.45 | 89.64 | 36.54 | 90.15 | 51.93 | 83.22 |
| FedLAPT | 80.67 | 55.75 | 82.44 | 67.51 | 64.67 | 78.53 | 86.48 | 55.97 |
| FOCoOp | 18.02 | 94.75 | 15.97 | 96.47 | 23.71 | 93.71 | 17.23 | 96.03 |

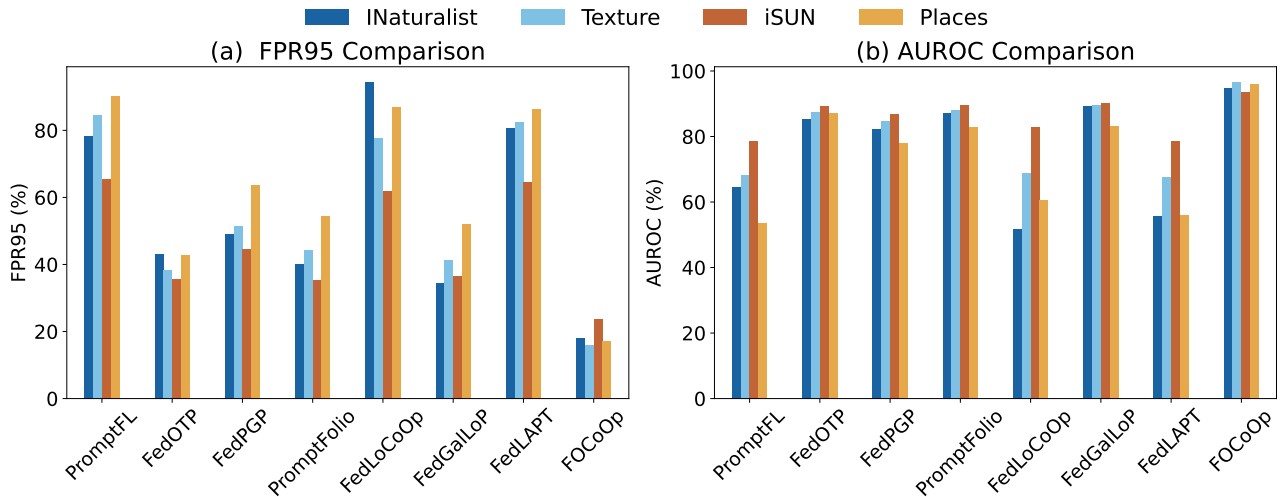

Figure 8: Detection Comparison on CIFAR-100.

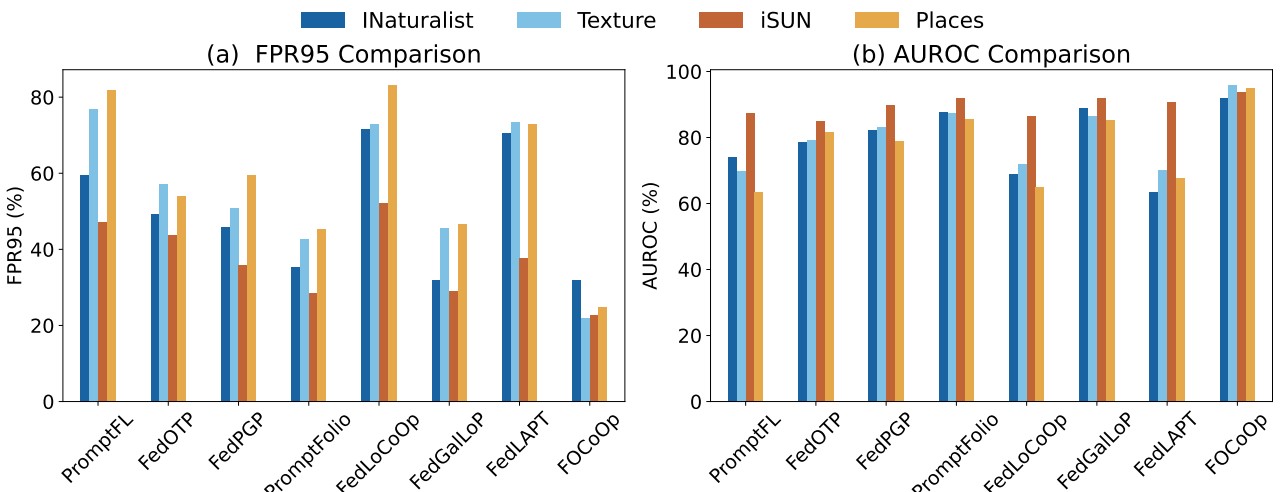

Figure 9: Detection comparison on TinyImageNet.

Table 12: Detection results on TinyImageNet non-overlap pathological heterogeneity.

| Dataset | INaturalist | | Texture | | iSUN | | Places | |
|---|---|---|---|---|---|---|---|---|
| Method | FPR95 | AUROC | FPR95 | AUROC | FPR95 | AUROC | FPR95 | AUROC |
| PromptFL | 59.64 | 74.22 | 76.75 | 69.82 | 47.25 | 87.45 | 81.94 | 63.56 |
| FedOTP | 49.36 | 78.48 | 57.12 | 79.18 | 43.66 | 84.84 | 54.07 | 81.52 |
| FedPGP | 45.76 | 82.13 | 50.86 | 83.05 | 35.76 | 89.84 | 59.63 | 79.00 |
| PromptFolio | 35.44 | 87.78 | 42.84 | 87.34 | 28.58 | 91.98 | 45.34 | 85.68 |
| FedLoCoOp | 71.62 | 68.96 | 72.96 | 72.07 | 52.17 | 86.49 | 83.06 | 64.96 |
| FedGalLoP | 32.01 | 88.96 | 45.54 | 86.53 | 28.97 | 92.04 | 46.66 | 85.23 |
| FedLAPT | 70.46 | 63.41 | 73.46 | 70.15 | 37.75 | 90.71 | 72.89 | 67.58 |
| FOCoOp | **31.84** | **91.98** | **21.98** | **95.75** | **22.61** | **93.80** | **24.82** | **94.99** |

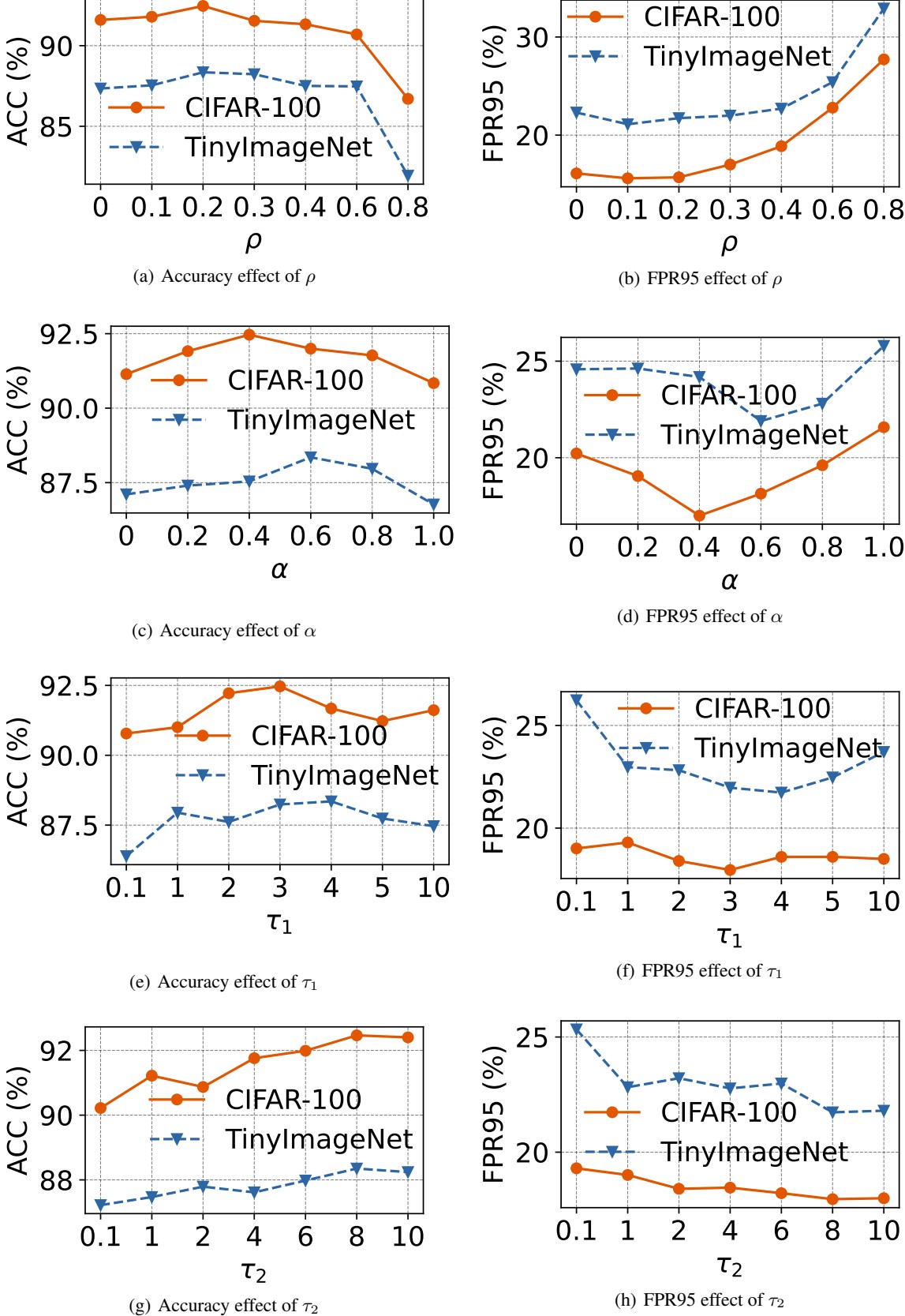

Figure 10: Hyperparameter sensitivity studies.

Table 13: ID-C generalization for FL methods trained on CIFAR-100.

| ID-C Type | Brightness | Fog | Glass Blur | Motion Blur | Snow | Contrast | Frost | Impulse Noise | Pixelate | Spatter | Defocus Blur | Gaussian Blur | JPEG Compression | Saturate | Speckle Noise | Elastic Transform | Gaussian Noise | Shot Noise | Zoom Blur | Average |
|---|---|---|---|---|---|---|---|---|---|---|---|---|---|---|---|---|---|---|---|---|
| PromptFL | 65.05 | 55.33 | 26.33 | 50.90 | 55.03 | 54.78 | 54.87 | 44.07 | 52.66 | 59.96 | 57.10 | 54.66 | 40.85 | 57.93 | 37.58 | 47.61 | 31.48 | 37.41 | 54.41 | 49.37 |
| FedOTP | 88.67 | 82.32 | 56.43 | 79.82 | 82.32 | 81.30 | 81.62 | 68.20 | 79.96 | 85.99 | 84.40 | 83.01 | 71.24 | 83.69 | 65.06 | 76.61 | 57.73 | 64.58 | 83.03 | 76.63 |
| FedPGP | 82.33 | 74.84 | 46.75 | 71.58 | 74.56 | 72.79 | 73.74 | 60.48 | 72.19 | 77.75 | 76.50 | 74.25 | 61.86 | 75.83 | 56.72 | 68.01 | 49.44 | 55.91 | 74.36 | 68.42 |
| PromptFolio | 82.33 | 74.84 | 46.75 | 71.58 | 74.56 | 72.79 | 73.74 | 60.48 | 72.19 | 77.75 | 76.50 | 74.25 | 61.86 | 75.83 | 56.72 | 68.01 | 49.44 | 55.91 | 74.36 | 68.42 |
| FedLoCoOp | 60.17 | 50.80 | 20.89 | 46.89 | 50.36 | 49.00 | 49.40 | 35.49 | 47.73 | 52.23 | 52.74 | 49.96 | 37.26 | 50.93 | 32.77 | 42.90 | 27.25 | 32.16 | 49.84 | 44.15 |
| FedGalLoP | 88.68 | 82.35 | 54.76 | 79.83 | 82.74 | 81.16 | 81.63 | 74.74 | 80.80 | 86.20 | 83.80 | 82.00 | 71.15 | 83.60 | 66.45 | 76.10 | 59.70 | 65.95 | 82.28 | 77.05 |
| FedLAPT | 48.47 | 42.55 | 14.23 | 33.41 | 41.34 | 43.25 | 41.12 | 19.19 | 35.14 | 41.89 | 41.46 | 39.16 | 28.01 | 41.34 | 22.78 | 30.87 | 18.26 | 23.00 | 38.50 | 33.89 |
| FOCoOp | **91.79** | **85.26** | **60.83** | **82.68** | **86.39** | **83.06** | **85.87** | 73.44 | **83.06** | **86.89** | **86.45** | **85.26** | **74.27** | **85.46** | **68.01** | **80.74** | **60.34** | **67.19** | **85.09** | **79.58** |

Table 14: ID-C generalization for FL methods trained on TinyImageNet.

| ID-C Type | Brightness | Fog | Glass Blur | Motion Blur | Snow | Contrast | Frost | Impulse Noise | Pixelate | Defocus Blur | JPEG Compression | Elastic Transform | Gaussian Noise | Shot Noise | Zoom Blur | Average |
|---|---|---|---|---|---|---|---|---|---|---|---|---|---|---|---|---|
| PromptFL | 59.38 | 47.41 | 28.30 | 51.35 | 45.69 | 28.64 | 50.61 | 32.16 | 49.59 | 46.96 | 53.80 | 50.23 | 34.40 | 39.97 | 48.34 | 44.46 |
| FedOTP | 78.68 | 68.35 | 52.51 | 73.27 | 68.64 | 48.76 | 71.45 | 51.98 | 71.99 | 70.10 | 74.84 | 70.96 | 57.67 | 63.58 | 70.83 | 66.24 |
| FedPGP | 76.29 | 64.56 | 45.41 | 69.36 | 63.17 | 42.93 | 67.88 | 49.55 | 67.84 | 64.84 | 72.04 | 67.55 | 52.53 | 58.81 | 66.04 | 61.92 |
| PromptFolio | 83.40 | 73.06 | 55.07 | 77.73 | 72.57 | 51.68 | 76.14 | 57.17 | 75.99 | 73.69 | 79.69 | 76.02 | 60.71 | 66.72 | 74.93 | 70.30 |
| FedLoCoOp | 52.05 | 40.55 | 22.89 | 44.62 | 39.49 | 23.77 | 43.48 | 24.90 | 41.63 | 40.05 | 46.60 | 43.58 | 28.26 | 33.57 | 41.55 | 37.80 |
| FedGallLoP | 82.55 | 72.08 | 53.60 | 76.52 | 71.72 | 51.29 | 75.38 | 60.11 | 75.23 | 72.74 | 77.89 | 74.33 | 61.30 | 67.26 | 73.47 | 69.70 |
| FedLAPT | 55.79 | 43.53 | 23.51 | 48.12 | 42.15 | 27.03 | 46.33 | 26.28 | 46.13 | 43.86 | 50.14 | 46.50 | 29.65 | 35.03 | 44.90 | 40.60 |
| FOCoOp | **85.00** | **74.68** | **57.70** | **79.67** | **76.45** | **53.29** | **79.02** | 54.70 | **77.96** | **74.71** | **80.34** | **77.52** | 60.27 | **67.26** | **76.47** | **71.67** |

