# OpenReview forum: "FOCoOp: Enhancing Out-of-Distribution Robustness in Federated Prompt Learning for Vision-Language Models"
_ICML.cc/2025/Conference — ICML 2025 poster_

### Official Review · Reviewer_p5vg · 2025-03-07

**Overall Recommendation:** 5

**Summary:**

This work focuses on enhancing Out-of-distribution (OOD) robustness for federated prompt learning on pretrained vision-language models. The authors propose a federated OOD-aware Context Optimization framework, i.e., FOCoOp, which contains two main modules, i.e., BOS and GOC. BOS not only enhances class-level matching between image and prompts corresponding the class, but also maintains distribution-level separation between ID and OOD data. GOC further enhances the consistency in OOD generalization and detection among all clients. The authors evaluate FOCoOp in extensive experiments, validating its effectiveness in generalization and robustness in OOD shifts.

**Claims And Evidence:**

Yes, the claims in the paper are verified with convincing empirical results, e.g., the motivation example in Fig. 1, and subsequent experiments in section 4.

**Essential References Not Discussed:**

The references are sufficient, no extra essential reference is needed to include.

**Experimental Designs Or Analyses:**

Yes, the experiment designs and analysis are solid and comprehensive for verifying the claims and method designs. The study includes rigorous evaluations on plenty of real-world datasets, ensuring both performance and OOD robustness. The detailed analysis, such as alation studies and hyper-parameter sensitivity studies are sufficient.

**Methods And Evaluation Criteria:**

Yes, both the proposed method and evaluation criteria are well-aligned with the problem at hand. The OOD robustness of federated prompt learning is a crucial challenge in real-world scenarios, and the proposed FOCoOp framework effectively addresses multiple types of OOD shifts simultaneously. Moreover, the evaluation criteria are directly relevant to the contributions of FOCoOp in enhancing OOD robustness, ensuring that the assessments meaningfully reflect the framework’s effectiveness.

**Other Comments Or Suggestions:**

See weakness

**Other Strengths And Weaknesses:**

Strengths:
S1. The novelty of considering OOD robustness for federated prompt learning is vital and potentially necessary for the era of foundation models. And the authors make contributions to it with both methodology and evaluations.
S2. The authors provide the correct and sound theoretical inductions for the proposed methods, i.e., the optimizations of Eq.(6) and Eq.(8).
S3. The proposed bi-level prompts distribution robust optimization is quite interesting and insightful. The authors constrain the perturbation of global prompts within optimal transport cost for generalization, while perturbing OOD prompts within unbalanced optimal transport cost for the awareness of open-world unseen outliers.
S4. The experiments are comprehensive in comparing both federated prompt learning methods and federated learning with Clip-based OOD detection methods.


Weakness:
W1. There are some unclear notations, e.g., (1) $c()$ function in Eq.(7), (2) $d^+$ and $d^-$ in Eq.(2) and right part of lines 239-252, and (3) typos in step 13 of Algorithm 2, the last $\epsilon_u^o$ is supposed to be $\boldsymbol{o}_{\epsilon_u^o}$.
W2. The authors introduce three set of prompts, ID global prompts, local prompts, and OOD prompts, but their relationships could be further explained for readability.
W3. Based on the empirical results, it is interesting to find that FOCoOp seems to have more contributions on OOD detection tasks while insignificant improvements on ID and ID-C generalization. The authors could provide more explanation with regarding to it.

**Questions For Authors:**

Can the authors explain the confusion, i.e., the local prompts are aggregated in lines 158-157, while global prompts are aggregated in lines 246-147?

**Relation To Broader Scientific Literature:**

This work builds on prior research in federated learning and vision-language models, aiming to enhancing OOD robustness. Unlike previous studies, FOCoOp optimizes prompts to enhance both class-level and distribution-level separations per client while maintaining consistency at the server, improving both performance and OOD robustness.

**Theoretical Claims:**

Yes, the theoretical claims and proofs are correct and reasonable.

---

> ### Author Rebuttal · Authors · 2025-03-25
>
> **1 We are willing to release our code and have made the project available at [https://anonymous.4open.science/r/FOCoOp/](https://anonymous.4open.science/r/FOCoOp/).  It will be included in the final version of the paper.**
>
> **2 $\mathfrak{c}()$ in Eq.(7) is the cost function, which can be computed via L2-norm distance.**
>
> **3 d+ and d- indicate the positive and negative distance set, respectively.**
> Specifically, $d^+$ means we compute the set by satisfying $\eta$ percentile of positive cosine distance, while $d^-$ means the set satisfying $\eta$ percentile of negative distance.
>
> **4 We will carefully revise our presentation and fix the typos.**
>
> **5. Clarification on the Role and Aggregation of Prompts.**
> As described in lines 78–81, the three sets of prompts serve distinct roles:
>
> **Global prompts** capture the shared ID global distribution across all clients.
>
>
> **Local prompts** capture each client's ID personalized but heterogeneous distribution.
>
>
> **OOD prompts** are trained to mismatch with ID data, preventing them from mistakenly aligning with samples that are unseen locally but present in other clients.
>
>
> **During training, each client sends both global prompts and OOD prompts to the server, while leaving local prompts at clients.**
> The server aggregates global prompts to learn a consistent alignment with the global distribution, and selects the most dissimilar OOD prompts to represent a consistent misalignment across clients.
> Since OOD prompts are shared and distribution-agnostic, they generalize well to open-world OOD datasets, leading to superior performance in OOD detection tasks.
> We will elaborate on these design choices more clearly in the final version.
>
>
> **6 Explanation on FOCoOp’s greater impact on OOD detection.**
> We appreciate the reviewer’s observation. **FOCoOp is the first method that is designed to enhance OOD robustness while maintaining strong performance on standard FPL tasks.** This is indeed reflected in the results: while improvements on ID and ID-C generalization are moderate (as existing FPL methods already achieve strong performance), FOCoOp demonstrates substantial gains in OOD detection.
> This discrepancy arises from our explicit design: FOCoOp introduces OOD prompts and uses a DRO objective to explore a broader prompt space. While this helps generalization to some extent, the major benefit lies in detection. Specifically, we distinguish between seemly OOD prompts (which match with ID samples that are unseen locally but seen globally) and strict OOD prompts (which are mismatched with all ID data), enabling more accurate identification of OOD data.
> We will clarify this distinction and its impact in future revisions.
>
>
> **7 Clarification on aggregating prompts.**
> In lines 157.left-158.left, we present a **general formulation of PFL** and refer to **prompts matched with local data on each client as local prompts**. As described in lines 246–247, we also explicitly introduce the method of **FOCoOp, which first aggregates global prompts across all clients**, and calibrates global prompts later by Eq. (10). We will clarify this distinction more clearly in the next version.

---

### Official Review · Reviewer_NsKV · 2025-03-11

**Overall Recommendation:** 4

**Summary:**

In this paper, the authors provide a Federated OOD-aware Context optimization framework named as FOCoOp. FOCoOp optimizes three sets of prompts for generalization, personalization, and detection, respectively. In the client local training, the authors devise a BOS module to maintain the class-level and distribution separations for discriminate client data. And a GOC module tackles challenges of enhancing the consistency of OOD robustness among clients.

**Claims And Evidence:**

The claims are supported by clear and convincing evidence. For example, Federated prompt learning methods are inferior in OOD robustness, which is evidently reflected in Fig. 1.

**Essential References Not Discussed:**

The essential references are discussed.

**Experimental Designs Or Analyses:**

The experimental design and analysis are well-structured and conclusive. Firstly, the authors select a diverse range of datasets for various tasks. Secondly, they evaluate FOCoOp and its representative baselines under different levels of data heterogeneity and varying client participation. Lastly, the analyses align with the claimed contributions of enhancing OOD robustness, supported by comprehensive experimental discussions.

**Methods And Evaluation Criteria:**

The proposed method and evaluation make sense. Specifically, FOCoOp aims to maintain the performance and enhance OOD robustness, both of which are evaluated valid in various empirical setups. The authors also provide ablation studies for the effect of BOS and GOC modules.

**Other Comments Or Suggestions:**

The authors should clarify some typos, i.e.,
a) T^g_c in line 262,
b) robustness3 in line 416,
c) \pho rather p in line 742,
d) two rather three in line 328,
e) last \epsilon in line 787.

**Other Strengths And Weaknesses:**

Strengths:
1.The paper is well organized and the main idea is easy to follow.
2.The problem of enhancing OOD robustness in federated prompt learning is vital and novel for the development of foundation models. This motivates border studies of this topic in the future.
3.The proposed methods are sound and effective in addressing challenges related to the improved discrimination of in-distribution classes, handling distribution shifts in clients, and ensuring consistent OOD robustness under client heterogeneities.
4.The experiments and analyses are extensive and investigable for contribution of model designs.
5.The designed crucial modules provide motivative insights. For example, the alignment between global prompts and OOD prompts are coupled with semi-unbalanced optimal transport, and further enhance the discrimination of two sets of prompts based on coupling results.


Weakness:
1.Though the good organization of main paper, the presentation should be refined, e.g., clarifying typos.
2.Some details need more discussions: a) the contents in supplementary materials, b) the data used for federated prompt learning and testing, and c) the choice of baseline methods.
3.The need to optimize multiple prompt sets (Global, Local, OOD) results in higher computational costs compared to traditional FPL methods.

**Questions For Authors:**

Q1: Are the OOD shifts data used for the optimization of prompts? If so, could the authors explain how these OOD shifts data are trained in the process. If not, please explain the objective of Eq.(1) for clarity.

**Relation To Broader Scientific Literature:**

The proposed FOCoOp contributes maintaining the performance and enhancing OOD robustness on federated prompt learning for vision-language models. It is quite novel and vital for the development of utilizing foundation models in a privacy-preserving and robustness-aware approach.

**Theoretical Claims:**

The proofs of theoretic claims are correct.

---

> ### Author Rebuttal · Authors · 2025-03-30
>
> **1 We will carefully refine current version and correct typos.**
>
> **2 The supplemental materials consist of five sections: (A) related work details, (B) algorithms, (C) theoretical analysis of optimization, (D) datasets and implementation, and (E) additional experimental results.**
> **In Section E**, we report the results on CIFAR-100 and TinyImageNet under different Dirichlet distributions (Tables 6–7), which are consistent with Tables 1–3 and further verify OOD robustness. Table 4 has been split into Tables 8 and 9 for clarity.
>  Tables 10–11 provide numerical results for Figure 4 (OOD detection on various OUT datasets), while Tables 12–13 are numerical results corresponding to Figure 5 (generalization on different ID-C datasets). Figures 8–9 are also enlarged for better readability. We will add them in a future version.
>
> **3 Clarification on data usage in FPL**
>
> **We do not use OOD data when optimizing prompts. Please refer to Answer 4 in Review zfft for further details.**
> As indicated in Eq. (1), the final evaluation targets minimizing classification errors on ID and ID-C test sets, and minimizing the failure to detect OUT data. We will clarify this more explicitly in the future. A summary of training and evaluation settings is provided below:
>
> **Training datasets:**
>
> CIFAR-100, TinyImageNet, Food101, DTD, Caltech101, Flowers, and OxfordPets, each split into train/test sets and simulated as heterogeneous clients using Pathological or Dirichlet partitioning.
>
> DomainNet and Office-Caltech10 use an N-1 domain split strategy.
>
>
> **Evaluation settings:**
>
> ID ACC: Accuracy on the test sets of the above training datasets.
>
>
> ID-C ACC: Accuracy on CIFAR-100-C, TinyImageNet-C, and the held-out domain in DomainNet and Office-Caltech10.
>
>
> OOD FPR95 / AUROC: Evaluated on Places365, Texture, iSUN, LSUN-C, and LSUN-R.
>
>
> **4 Baseline Selection.**
> We select baselines from the state-of-the-art (SOTA) personalized federated learning (PFL) methods and centralized prompt-based OOD detection methods. For centralized prompt-based OOD baselines, we choose GalLop and LAPT due to their strong performance in both generalization and OOD detection.
> We also include LoCoOp, as GalLop is proposed as an improvement over it.
>
> **5 Our computation cost is comparable to existing FPL methods such as PromptFolio and FedOTP, with the global and local prompt computation cost being $\mathcal{O}(KEBCd)$. The introduction of OOD prompts adds an extra but manageable overhead of $\mathcal{O}(KEB(C+U)d)$, where $U$ is the number of OOD prompts.** Since $U$ is a controllable parameter, incorporating OOD prompts is a reasonable choice for significantly improving OOD detection performance.
> All notations are defined in our main paper, i.e., client numbers $K$, local epochs $E$, averaged batch size $B$, class number $C$, and prompt embedding $d$.
> Furthermore, we measured the average computation time for each communication round. An extra set of OOD prompts only increases computation cost by 4.76%, and 1.61% for CIFAR100 and TinyImageNet. The most computational burdens stem from encoding features rather than prompt learning. This indicates that employing three sets of prompts remains a practical and efficient design. FOCoOp containing BOS and GOC causes slightly larger but still controllable computation, which increases cost by 1.58s(9.06%) and 1.64s(6.18%) for CIFAR100 and TinyImageNet, respectively.
> | Method| CIFAR100(s) | Increase | TinyImagenet (s) | Increase |
> |-|-|-|-|-|
> | PromptFolio| 17.4765| -| 26.5969 | - |
> | FOCoOp-w/o-dro&uot | 18.3078 | +0.8313 (+4.76%) | 27.0261           | +0.4292 (+1.61%)    |
> | FOCoOp  | 19.0597 | +1.5832 (+9.06%)  | 28.2402| +1.6433 (+6.18%)    |
>
>
> **6 We will carefully refine our presentation and correct typos.**

---

> > ### Comment · Reviewer_NsKV · 2025-04-03
> >
> > The authors provided more detailed explanations regarding the presentation, data usage, and time complexity. These clarifications are reasonable and address most of my concerns, so I am raising my rating.

---

### Official Review · Reviewer_zfft · 2025-03-13

**Overall Recommendation:** 3

**Summary:**

Federated Prompt Learning (FPL) allows models to adapt across clients while maintaining data privacy. However, current methods face challenges balancing performance and robustness, especially when encountering out-of-distribution (OOD) data shifts, limiting their real-world reliability. This is mainly due to data heterogeneity among different clients. To overcome these challenges, FOCoOp introduces a strategy utilizing three types of prompts to create clear separations at both class and distribution levels. Experiments demonstrate that FOCoOp effectively improves robustness and adaptability to OOD scenarios in heterogeneous federated learning settings.

**Claims And Evidence:**

1. In lines 68-72, the authors mention that "The crucial reason is that each client maintains local OOD robustness on heterogeneous data distribution, which is inconsistent among FPL." It is unclear why the maintenance of local OOD robustness on heterogeneous data distributions would be considered inconsistent in FPL, especially given that prompts are able to handle other downstream tasks, like different data distributions. Could the authors elaborate on this inconsistency with a more detailed explanation or provide empirical results that clarify this aspect?

**Essential References Not Discussed:**

No.

**Experimental Designs Or Analyses:**

1. Could the authors provide any experimental results demonstrating that FOcoOp is effective in exploring OOD data?

2. Are there any experiments that show how the proposed three types of prompts create clear separations at both the class and distribution levels?

3. Do the proposed methods work with ResNet CLIP models? Given that ViT-B/16 may be difficult for clients to deploy, it would be helpful to understand if the methods are also effective with more accessible models like ResNet-50.

**Methods And Evaluation Criteria:**

1. In lines 234-235, the authors mention that "The OOD prompts in client match the unseen data from other clients with high similarity scores, hurting the generalization in global view." Could the authors clarify why these OOD prompts would match unseen data from other clients? Is this a general case or a specific situation? A more detailed explanation would help in understanding the reasoning behind this claim.

2. In lines 250-254, the authors mention that "The OOD prompts capture the misalignment between local ID data and prompt context, mistakenly identifying the ID data from other clients as outlier." Could the authors clarify why the OOD prompts would capture this misalignment between local ID data and the prompt context? A more detailed explanation or simple results to support this claim would be appreciated.

**Other Comments Or Suggestions:**

A typo: in line 81-82 in Section 2.1, two CLIP2FL methods.

**Other Strengths And Weaknesses:**

1. What are the privacy considerations regarding the client prompts, particularly when they are transmitted to the server? Could the authors address how this aspect is handled or discuss it with any related works?

**Questions For Authors:**

Please refer to the above sections.

**Relation To Broader Scientific Literature:**

The paper proposes FOCoOp, a Federated OOD-aware Context Optimization framework to enhance robustness and performance in Federated Prompt Learning (FPL) for VLMs.

**Theoretical Claims:**

Theorems 3.1 and 3.2.

---

> ### Author Rebuttal · Authors · 2025-03-30
>
> **1 Inconsistency in maintaining local OOD robustness in FPL.**
>
> **FOCoOp aims to improve the OOD robustness of FPL concerning the global distribution, which covers all clients' training data.** The consistency we focus on lies in detecting semantic shifts beyond all client data and ensuring generalization under covariate shift and client heterogeneity within the global distribution.
>
> **However, this consistency cannot be directly achieved by prompt tuning for local robustness.**
> While prompts can transfer the knowledge of foundation models, they are adapted to heterogeneous local data during training, **leading to inconsistent matching patterns across clients.**
> This makes prompts wrongly identify any data different from local distribution as OUT, degrading both generalization and detection.
> **Results in Tab. 1-3 reflect this limitation.**
> PromptFL and FedLoCoOp lack consistency mechanisms and underperform in all metrics.
> Similarly, FPL methods like FedOTP and PromptFolio aim to balance generalization and personalization, yet still fail in detection.
>
> **2 Reason for OOD prompts in client match unseen data from other clients.**
>
> **This is a general case caused by heterogeneous data distributions across clients.** In federated settings, clients often have imbalanced or missing classes, e.g., client 1 may only have data of dog and cat classes, while client 2 has dog, cat, and fish.
> During training, OOD prompts are optimized to avoid resembling locally seen data. However, lacking the reference of global distribution, OOD prompts may mistakenly match with samples locally unseen but present in other clients. For instance, in client 1, OOD prompts may falsely match "fish" images as OUT, but fish images are in client 2.
>
> **3 We realize the goal of OOD prompts, i.e., capturing misalignment between local ID data and the OOD prompt context, by explicitly optimizing Eq. (5).** Specifically, we penalize higher similarities between OOD prompts and local ID data(Eq. 3), and ensure the total similarity of OOD prompts is lower than that of ID prompts(Eq. 4). **This design notably improves OOD detection across all evaluated tasks in Sec 4.2.** For example, it consistently improves the detection of different semantic shift datasets in Fig.4.
> However, since each client only observes a subset of the global distribution, the learned OOD prompts may mistakenly match ID samples unseen in this client but valid from other clients as OUT. Therefore, the GOC module is used to enhance consistent OOD robustness across clients.
>
> **4 Experiments on exploring OOD data.**
> We clarify that **only in-distribution (ID) data is used during training**, which aligns with federated settings where OOD data is typically unavailable. **Our method does not explicitly explore OOD data, instead, it uses a DRO-based objective to explore prompt space**, allowing global (OOD) prompts to better match (mismatch) ID samples.
>
> **All OOD data (both ID-C and OUT) are used only for evaluation(main paper Table 5), not for training prompts.**
>
> Next, we evaluate OOD robustness on various OOD datasets, i.e., (1) Tables 1–2 show FOCoOp’s strong generalization to ID-C datasets (e.g., CIFAR100-C, TinyImageNet-C) and its improved detection of OUT data (e.g., Texture), (2) Extra evaluation of various OUT datasets also verified that FOCoOp can consistently enhance OOD robustness. These confirm the effects of exploring prompts in FOCoOp.
>
> **5 We can verify the clear separations for both class and distribution levels by similarity matrix.** In **[link](https://anonymous.4open.science/r/FOCoOp/prompt_separation.png)**, we model FOCoOp on Cifar10 (10 ID prompts and OOD prompts), and sample 100 images per class to compute the average of similarities between images and prompts. The diagonal of the ID prompt matrix shows the highest similarities, suggesting intra-class alignment and clear class separation. Meanwhile, the similarities of OOD prompts are notably lower than those of ID prompts, further indicating clear distribution separation.
>
> **6 We extend our evaluation to ResNet50 on CIFAR100 and TinyImageNet. The results validate that FOCoOp maintains strong generalization and detection capabilities even with smaller models.**
>
> Pathological Non-overlap (10 clients, K=10)
> |Dataset|||CIFAR100|||TinyImageNet||||
> |-|-|-|-|-|-|-|-|-|-|
> |Method(%)|ACC|CACC|FPR95|AUCROC|ACC|CACC|FPR95|AUCROC|
> |PromptFolio|51.96|47.47|50.14|86.45|44.70|30.64|57.57|83.03|
> |FedOTP|55.34|50.98|61.38|74.92|43.61|28.98|72.67|73.86|
> |FedLoCoOp|17.03|12.09|93.36|52.66|9.44|4.96|93.63|56.40|
> |FOCoOp|60.94|55.92|28.83|95.54|49.96|35.89|53.51|87.75|
>
> **7 Privacy analysis. Prompts encode only class-level statistics rather than instance-specific information, making it non-trivial to infer original individual image data stored on clients.** The privacy of class-level statistics can be strengthened by applying differential privacy on prompts before sending to server.
>
> 8 We will carefully fix the typos mentioned.

---

### Official Review · Reviewer_cE28 · 2025-03-13

**Overall Recommendation:** 4

**Summary:**

The paper focuses to devise an out-of-distribution (OOD) enabled federated prompt learning method based on CLIP model. In addition to global and local prompts, the papers proposes to learn OOD prompts to enable OOD detection at each client in the federated learning framework. Similarity scores in the CLIP embedding space are used to optimize all the three prompts, and further encourage to make the prediction probability of local and global prompts to be larger than OOD prompts on the client data. One of the main contributions of the paper is to introduce Bi-level Prompts Distribution Robust Optimization (BDRO), that perturbs both global and OOD prompts to the worst case at each client using optimal transport discrepancy to enable wider distribution exploration. Besides, the paper also provides an understanding that OOD prompts of one client could observe data from other client as OOD. To eliminate such discrepancy, the paper extracts relevant global prompt information from such OOD prompts and extract strictly OOD prompts that are consistent across all the clients using semi-unbalanced optimal transport. Experiments conducted on CIFAR-100, TinyImageNet and their corrupted variants show better clean accuracy and generalization with significantly improved OOD scores compared to existing federated learning approaches. Similar improvements are also noticed across datasets like Food101, DTD, Caltech101, Flowers102 and OxfordPets. Domain generalization on datasets like DomainNet and Office show competetive results.

## update after rebuttal
I thank the authors for providing sufficient justification and explanation addressing my questions. I encourage the authors to reflect this rebuttal in their revised version and make it easy for the readers to comprehend the method. Given the current state of the paper and based on the author response, I am inclined to accept the paper.

**Claims And Evidence:**

The claims on improving the performance with bi-level distribution robust optimization of both global and OOD prompts and also with  global-view OOD consistency across clients are sufficiently backed up by the empirical evidence shown in Tables 1, 2 and 3.

**Essential References Not Discussed:**

To my knowledge, I see that authors have cited and discussed the relevant literature.

**Experimental Designs Or Analyses:**

The authors follow the existing literature experimental protocol setting to conduct their experiments.

**Methods And Evaluation Criteria:**

Evalution Criteria: To my knowledge, authors made an exhaustive evaluation with various benchmark datasets like CIFAR-100, TinyImageNet, Food101, DTD, Caltech101, Flowers102 and OxfordPets, DomainNet and Office.

Methods: Both the bi-level distribution robust optimization and global-view OOD consistency are heavily developed based on the optimal transport. Despite the empirical improvements, it is not clear to realize the why optimal transport is right fit in this setting. In particular, authors intend to create worst case global and OOD prompts via bi-level distribution robust optimization. The rationale to create worst case prompts and the choice to make use of optimal transport to achieve it is not clear and justified. The authors connect it to wider distribution exploration, but it is hard to realize how such exploration is possible with the proposed setup. Moreover, in equation (7), the \hat{t}^g gets sampled from \hat{P} and \hat{t}^o gets sampled from \hat{Q}. How do these \hat{P} and \hat{Q} are defined, as these are not clear even after looking into Algorithm 2 in appendix. Main paper should contain insights about them. There is a function that operates on (\hat{t}^g, t^g), there is no definition given to this function. Similarly, the equation 8 related to global-view OOD consistency, there is no clear explanation on computing or obtaining seemly OOD prompts and strict OOD prompts using optimal transport. Overall, the current explanation of the method needs a good refinement. A detailed explanation supporting the equations would provide a good understanding to the readers.

**Other Comments Or Suggestions:**

None

**Other Strengths And Weaknesses:**

Strengths:
1. The paper show significant improvements for OOD detection while improving accuracy on federated prompt learning setting across multiple benchmarks.
2. Ablation studies provide better understanding of the method.
3. I also appreciate the authors for providing the Algorithms and details on their experimental setup.

Weakness:
1. In Figure 2, the server block is hard to infer based on the illustration. Explanation in the caption could help to understand.
2. The explanation of the method section lacks clarity and justification, and requires refinement to improve readability and understanding the concepts presented in the section. Please see my comments in the method block above.

**Questions For Authors:**

What is the rationale to create worst case prompts?
What makes the optimal transport the right choice to make use of for the problem?
How do the \hat{P} and \hat{Q} are defined?
What is the function definiton that operates on (\hat{t}^g, t^g) in equation 7?

Based on my comments on the method section, please incorporate the suggestions and improve the readability. The results shown in the paper are promising, but it should be complemented with a well explained method section.

**Relation To Broader Scientific Literature:**

There is a list of literature focusing on federated prompt learning methods using CLIP model. This work complements this literature by introducing OOD detection into the framework that could be leveraged by all the participating clients.

**Theoretical Claims:**

There are two theorems presented in the paper. I have not verified their correctness.

---

> ### Author Rebuttal · Authors · 2025-03-30
>
> **1Reasons for using optimal transport (OT).**
>
> **OT is a powerful tool for comparing distributions, and is used for different goals in BOS and GOC.**
>
> **In BOS, OT is used to constrain uncertainty set in distributionally robust optimization (DRO).** Unlike KL-divergence, capturing categorical distribution without considering geometry in feature space, OT divergence preserves the geometry of latent feature spaces, which is vital to text-image feature matching in VLMs. As noted in lines 211.left-188.right, global prompts enhance robustness in ID data in all clients and covariate-shift data near ID data, making OT as a suitable constraint. However, we adopt unbalanced OT (UOT) to constrain the uncertainty set of OOD prompts. Because it combines the strengths of OT and KL-divergence, capturing both geometric and non-geometric semantic shifts.
> **In GOC, it uses semi-UOT to improve discrimination between global prompts and OOD prompts.** By coupling global prompts and all OOD prompts, we can select seemly OOD prompts based on the close distance, and distribute mostly distant (strict) OOD prompts to all clients.
>
> **2 The reason for creating worst-case prompts and how to realize wider distribution exploration.**
>
> We obtain the worst-case prompts via DRO, as defined in Eq. (6). Specifically, we construct an uncertainty set $P$ for global prompts, which is centered around the original distribution of global prompts $P_0$​, and bounded by an OT divergence threshold $\eta_1​$. Within this set, we identify and optimize the perturbed prompts that yield the worst performance, as formulated in Eq. (6). The distribution of worst-case prompts is denoted as $\hat{P}$.
> The same procedure with UOT-divergence is applied to OOD prompts, where original distribution $Q_0$, uncertainty set $Q$, and worst-case prompts $\hat{\boldsymbol{t}}^o \sim \hat{Q}$ are corresponding notations.
>
> Since we match images with textual prompts that yield the worst performance, the matching process is no longer point-to-point. Instead, it becomes a point-to-uncertainty-set matching, enabling more robust alignment in a wider semantic space.
>
>
>
>
>
> **3 In Eq.(7), $\mathfrak{c}(\hat{t}^g,t^g)$ is the cost function of optimal transport, we implement it by computing L2-norm distance.**
>
> **4 Explanation on computing Eq.(8) to get seemly and strict OOD prompts.**
> As in line 270.left, we compute all costs between aggregated global prompts and OOD prompts from all clients, and we regularize the assignments for OOD prompts with soft KL regularization while the assignments for global prompts are balanced regularization. Then we solve the mapping $\pi$ by theorem 3.2, where we use ${\pi^*}^\top 1_{\text{KU}}$ as distance. Based on it, we can rank OOD prompts, and the seemly OOD prompts are close to global prompts, but strict ones are mostly distant. The whole procedure can be found in lines 266.left-265.right.
>
> **5 We will add explanations for server block in Fig. 2.** Specifically, in global-view OOD consistency, we use Semi-UOT to couple aggregated global prompts and all client OOD prompts. Then we take  ${\pi^*}^\top 1_{\text{KU}}$ as probability/distance, to select seemly OOD prompts and strict OOD prompts for the next communication round. Based on mapping, we can re-assign global prompts with seemly OOD prompts by Eq.(10), and remain strict OOD prompts by Eq.(11). Then server sends the final global prompts and OOD prompts to all clients for consistency.
>
> **6 We sincerely appreciate the reviewer’s constructive suggestions, and will improve the clarity of our revised version.**

---

### Decision · Program_Chairs · 2025-05-01

**Decision:**

Accept (poster)

**Comment:**

Overall, the reviewing panel is very positive with this paper. We all find this paper acceptable.

The rebuttal has also resolved multiple concerns effectively, reinforcing its acceptability.

I also have a set of questions for the authors at the beginning of the reviewer-author discussion phase. My main concerns are that the authors have missed two relevant works on federated prompt-tuning and federated LoRA.

The work on federated prompt-tuning could potentially be used for the same setting while the federated LoRA work proposes an orthogonal approach to federated fine-tuning (though not specifically designed for OOD).

[1] https://arxiv.org/abs/2409.05976 (NeurIPS-24)

[2] https://arxiv.org/abs/2502.19752 (NeurIPS-24)

However, with clarification from authors, the proposed approach has an added benefit/advantage of being able to tune text prompts to visual data while the above use vision-prompt for visual data; or text prompt for text data.

Given this, I consider those as concurrent approaches. The authors are encouraged to cite and position their work with the above in the camera-ready paper as they have done comprehensively in their response to me.

--

In summary, this is a solid work. I recommend acceptance.